# FUS-dependent loading of SUV39H1 to *OCT4* pseudogene-lncRNA programs a silencing complex with *OCT4* promoter specificity

Michele Scarola[1,2], Elisa Comisso[1,2], Massimo Rosso[1,3], Giannino Del Sal [1,3], Claudio Schneider[1,2], Stefan Schoeftner [1,3 ✉] & Roberta Benetti [1,2 ✉]

The resurrection of pseudogenes during evolution produced lncRNAs with new biological function. Here we show that pseudogene-evolution created an *Oct4* pseudogene lncRNA that is able to direct epigenetic silencing of the parental *Oct4* gene via a 2-step, lncRNA dependent mechanism. The murine *Oct4* pseudogene 4 (*mOct4P4*) lncRNA recruits the RNA binding protein FUS to allow the binding of the SUV39H1 HMTase to a defined *mOct4P4* lncRNA sequence element. The *mOct4P4*-FUS-SUV39H1 silencing complex holds target site specificity for the parental *Oct4* promoter and interference with individual components results in loss of *Oct4* silencing. SUV39H1 and FUS do not bind parental *Oct4* mRNA, confirming the acquisition of a new biological function by the *mOct4P4* lncRNA. Importantly, all features of *mOct4P4* function are recapitulated by the human *hOCT4P3* pseudogene lncRNA, indicating evolutionary conservation. Our data highlight the biological relevance of rapidly evolving lncRNAs that infiltrate into central epigenetic regulatory circuits in vertebrate cells.

[1] Laboratorio Nazionale—Consorzio Interuniversitario per le Biotecnologie, Laboratorio Nazionale (LNCIB), Padriciano 99, 34149 Trieste, Italy. [2] Dipartimento di Area Medica (DAME), Università degli Studi di Udine, p.le Kolbe 4, 33100 Udine, Italy. [3] Dipartimento di Science della Vita, Università degli Studi di Trieste, Via E. Weiss 2, 34127 Trieste, Italy. ✉email: sschoeftner@units.it; roberta.benetti@uniud.it

Pseudogenes are non-functional gene copies that have lost protein coding potential. Precise annotation and integration of functional genomics data revealed a high number of pseudogenes that have evolved to new functional elements, producing long noncoding RNAs (lncRNAs) in a tightly controlled manner[1,2]. In many cases, sequence similarity of pseudogene derived lncRNAs with parental gene transcripts provides the rational basis for pseudogene dependent control of ancestral gene expression. Pseudogene lncRNAs have been reported to compete with parental gene transcripts for miRNAs or RNA binding proteins or, alternatively, can give rise to endo-siRNAs[3–8]. Antisense transcription of pseudogenes can mediate epigenetic silencing of ancestral genes in trans, presumably by pairing with ancestral sense gene transcripts[9,10]. Remarkably, pseudogene derived lncRNAs have also been demonstrated to act as scaffold for chromatin modifying complexes that can modulate gene expression at multiple loci across the genome[11,12].

The transcription factor OCT4 is central for vertebrate embryonic stem cell (ESC) pluripotency and cancer cell biology and represents a hallmark model for the multifaceted pathways of pseudogene lncRNA mediated regulation of parental gene expression[10,13–23]. During evolution, the murine and human *Pou5f1/POU5F1* genes, that encode OCT4, gave rise to five processed murine (*Pou5F1P1–Pou5F1P5*) and eight processed human pseudogenes (*POU5F1P1–POU5F1P8*), with validated lncRNA expression[17,24–26]. Hereinafter, *Pou5f1* and *POU5F1* pseudogenes will be referred to as *Oct4/OCT4* pseudogenes. Murine *Oct4* pseudogene derived lncRNAs show defined pattern of expression during mouse embryonic stem cells (mESC) differentiation and specific cytoplasmic or nuclear localization, supporting evidence for the acquisition of new biological function[17]. In line with this, human *OCT4 pseudogene 4 and 5* lncRNAs alter ancestral gene expression by acting as classic ceRNAs, and pairing of the murine *Oct4-pseudogene 5* antisense lncRNA with *Oct4* transcripts has a role in guiding the histone methyltransferase (HMTase) EZH2 to the *OCT4* promoter[10,16,27].

We recently reported on a new mechanism of ancestral gene regulation that depends on pseudogene lncRNA dependent recruitment of an epigenetic silencing complex to the *Oct4* promoter in trans[17]. Induction of mESC differentiation results in efficient upregulation of the X-linked *mOct4P4* gene that encodes the *mOct4P4* lncRNA. The resulting nuclear restricted *mOct4P4* lncRNA forms a complex with the HMTase SUV39h1 and targets H3K9me3 and HP1 to the promoter of the parental *Oct4* gene on chromosome 17, leading to gene silencing in trans. Importantly, this mechanism does not involve pairing of *Oct4* sense and pseudogene antisense RNAs. To this end, lncRNA sequence determinants and evolutionary importance for *mOct4P4* pseudogene lncRNA dependent silencing of *Oct4* are not known.

Here, we show that the human *POU5F1P3* pseudogene derived lncRNA, *hOCT4P3*, is a functional homolog of the murine *Pou5f1P4* lncRNA in OVCAR-3 ovarian cancer cells, demonstrating evolutionary constraint on pseudogene–lncRNA-mediated epigenetic silencing of *OCT4*. Performing *mOct4P4* lncRNA pulldown experiments and a *mOct4P4* lncRNA deletion analysis we demonstrate that the RNA binding protein FUS and a 200 nucleotide *mOct4P4/hOCT4P3* region are essential for *Oct4/OCT4* silencing in mouse and human cells. Binding of FUS to endogenous, full length *mOct4P4/hOCT4P3* lncRNAs allows subsequent binding of SUV39H1 to the 200-nucleotide lncRNA element, forming a silencing complex with target specificity for the parental Oct4/OCT4 promoter. In experimental cell lines, the 200nt *mOct4P4/hOCT4P3* lncRNA sequence element is sufficient to guide SUV39H1 dependent *Oct4/OCT4* silencing, even in the absence of FUS.

We thus propose a model where FUS represents a licensing factor that mediates the accessibility of the 200 nucleotide *mOct4P4/hOCT4P3* to SUV39H1 binding, thereby imposing target specificity of the silencing complex towards the parental *Oct4/OCT4* gene promoter. Our data highlight the evolutionary relevance of pseudogene lncRNA mediated control of parental gene expression and the role of FUS in instructing the formation of an epigenetic regulatory complex with target site specificity defined by a lncRNA component.

## Results

**Conserved role of *hOCT4P3* and *mOct4P4* in silencing parental gene expression.** We recently demonstrated that the mouse *mOct4P4* lncRNA–SUV39H1 complex targets conserved promoter elements of the ancestral *Oct4* gene in trans, mediating gene silencing during mESC differentiation. To support the relevance of pseudogene lncRNA mediated epigenetic regulation of parental gene expression we tested whether this mechanism is conserved in human cells. To date, eight human *POU5F1* pseudogenes have been annotated in the human genome[25]. Similar to *mOct4P4*, the human *hOCT4P1*, *hOCT4P3*, and *hOCT4P4* pseudogenes have an exon structure that is similar to the *OCT4* mRNA and show 81%, 82%, and 82% overall sequence identity to *OCT4*, respectively[25]. We previously showed that OCT4 is frequently expressed in ovarian cancer cell lines and controls cancer relevant pathways in OVCAR-3 cells[15]. This identifies OVCAR-3 ovarian cancer cells as ideal model system to validate conservation of pseudogene lncRNA mediated silencing of parental OCT4. *hOCT4P3* lncRNA displays high sequence similarity to *mOct4P4* and reproduces nuclear localization pattern in a series of human ovarian cancer cell lines (Fig. 1a, b)[25].

Stable overexpression of *hOCT4P3* in OVCAR-3 cells leads to reduced OCT4 expression and downregulation of the self-renewal transcription factors SOX2, NANOG, and KLF4, indicative for impaired self-renewal circuits (Fig. 1c). Quantitative real-time polymerase chain reaction (RT-PCR) experiments revealed that *hOCT4P3* and *OCT4* transcript levels are 130- or 150-fold lower than the housekeeping gene *DAXX*. This indicates that, although present at low copy number, *hOCT4P3* has an important role in parental gene expression control (Supplementary Fig. 1a). To demonstrate conservation of *hOCT4P3* and *mOct4P4* function we used the CRISPR/dCas9–HAKRAB system to silence *hOCT4P3* or *mOct4P4* lncRNA expression in OCVAR-3 or mESC cells, respectively.

We first generated mESC and human OVCAR-3 ovarian cancer cell lines stably expressing an HA-tagged version of a catalytically dead Cas9 version fused to the Kruppel associated box (dCas9-HAKRAB; dCas9 empty cells). In a subsequent step dCAS9 empty cells were stably transfected with an expression vector encoding short-guide RNAs (sgRNAs) that locate dCas9–HAKRAB to the promoter region of the *Pou5f1P4/POU5F1P3* genes (dCAS9 sg*Oct4P4* mESCs or dCAS9 sg*OCT4P3* OVCAR-3 cells). Expression of dCAS9-HAKRAB and respective sgRNAs in experimental mESCs and OVCAR-3 cells was validated by western blotting and RT-PCR (Fig. 1d). We previously demonstrated that *mOct4P4* is efficiently upregulated during in vitro mESC differentiation[17]. Here, we used embryoid body (EB) differentiation as model system to address the impact of reduced *mOct4P4* lncRNA expression on self-renewal and early differentiation markers. dCAS9 empty and dCAS9 sgOct4P4 mESCs were cultivated in hanging drop cultures in the absence of the self-renewal factor leukemia inhibitory factor (see "Methods"). We found that upregulation of *mOct4P4* expression was strongly impaired during EB differentiation of dCAS9 sg*Oct4P4* mESCs (Fig. 1e). This effect was paralleled by inefficient *Oct4/OCT4* silencing during 10 days of EB differentiation on the RNA and protein

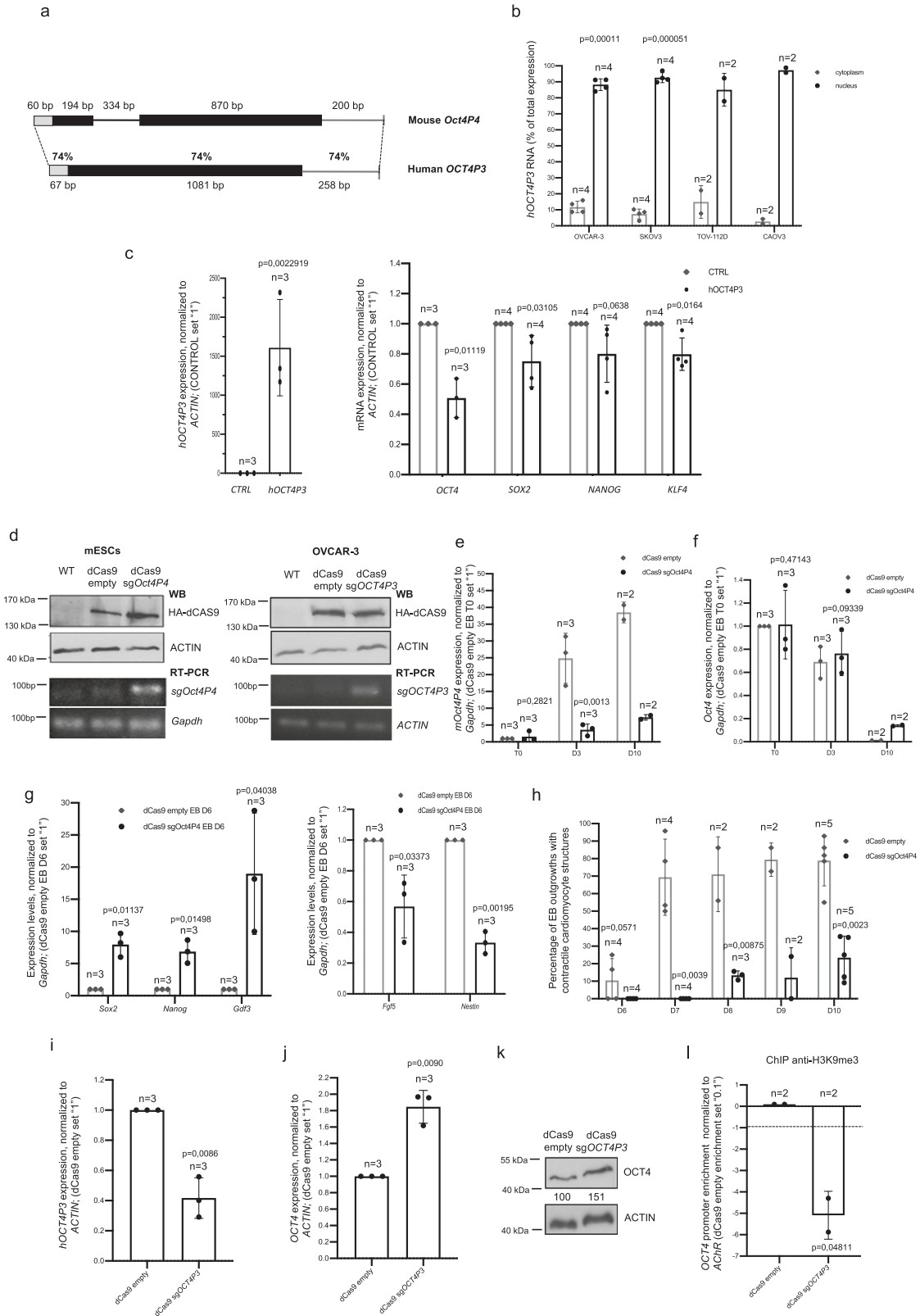

level (Fig. 1f, Supplementary Fig. 1b). Accordingly, we found increased expression of self-renewal transcription factors *Sox2*, *Nanog*, and *Gdf3* and reduced expression of early differentiation markers *Fgf5* and *Nestin* (Fig. 1g). On the functional level, dCAS9 sg*Oct4P4* embryoid bodies showed poor formation of contractile cardiomyocyte structures, indicative for in vitro differentiation defects (Fig. 1h, Supplementary Fig. 1c, Supplementary

Movies 1 and 2). Importantly, reduced expression of human *hOCT4P3* in dCAS9 sg*OCT4P3* OVCAR-3 cells was paralleled by increased expression of OCT4 at the RNA and protein level (Fig. 1j, k). This effect was paralleled by reduced H3K9me3 at conserved elements at the promoter of the parental *OCT4* gene (Fig. 1l). Based on our loss and gain of function experiment, we conclude that *hOCT4P3* recapitulates *mOct4P4* function in human

**Fig. 1 Conserved function of *hOCT4P3* and *mOct4P4* lncRNAs. a** Schematic representation of murine *mOct4P4* and human *hOCT4P3* pseudogenes. Length of sequence elements and percentage of sequence homology are indicated. Gray boxes, sequences with homology to *Oct4/OCT4* 5'UTR; gray lines, sequences with homology to *Oct4/OCT4* 3'UTR. A centrally located, 334-bp spliced fragment is exclusively present in *mOct4P4* (29). **b** Subcellular localization of *hOCT4P3* in human Ovarian Cancer cell lines OVCAR-3, SKOV3, TOV-112D, and CAOV3 as determined by quantitative RT-PCR (qRT-PCR). Shown values refer to the percentage of total RNA expression. **c** Quantitative RT-PCR analysis of *hOCT4P3* (left panel), *OCT4* and pluripotency marker genes (right panel) in OVCAR-3 cells stably expressing *hOCT4P3*. Expression levels were normalized against *ACTIN*. **d** dCas9-HA-KRAB western blotting analysis (top) and RT PCR analysis (bottom) of *Oct4* pseudogene guide RNA (sg*Oct4P4*, sg*OCT4P3*) in mouse embryonic stem cells (mESCs) (left panel) and OVCAR-3 cells (right panel). *ACTIN* and *Gapdh* were used as control. **e, f** *mOct4P4* lncRNA (**e**) and *Oct4* (**f**) expression in self-renewing mESCs (EB T0) and during 10 days of embryoid body (EB) differentiation (EB D3–D10). Expression levels were normalized to *Gapdh*. **g** qRT-PCR analysis of self-renewal marker genes (left panel) or markers of early mESC differentiation (right panel) in dCas9/sg*Oct4P4* mESCs. Expression values were normalized against gapdh. **h** Percentage of contractile cardiomyocyte structures in embryoid bodies (EBs) obtained from dCas9 or dCas9/sg*Oct4P4* cells. **i, j** Quantitative RT-PCR showing *hOCT4P3* lncRNA (**i**) and OCT4 (**j**) expression in dCas9 or dCas9/sg*OCT4P3* OVCAR-3 cells. Expression values were normalized using *ACTIN*. **k** OCT4 expression in knockdown dCas9 and dCas9/sgOCT4P3 OVCAR-3 cells as determined by western blotting. ACTIN was used as control. Numbers represent OCT4/ACTIN ratio (dCAS9 empty was set "100"). **l** Chromatin immunoprecipitation (ChIP) analysis on the *OCT4* promoter region in dCas9 and dCas9/sg*OCT4P3* OVCAR-3 cells using H3K9me3 antibodies. Error bars represent standard deviation; Precise *p* values are indicated; *n* number of independent experiments carried out.

OVCAR-3 cells. Importantly, data from dCAS9–HAKRAB loss of function models also demonstrate that endogenous *mOCT4P4* and *hOCT4P3* lncRNAs have a suppressive action on the *Oct4/OCT4* promoter in mESCs and OVCAR-3 cells.

Our results demonstrate the evolutionary conservation of H3K9me3 dependent silencing of parental *Oct4/OCT4* by mouse and human *mOct4P4* and *hOCT4P3* sense lncRNAs. This further implies the existence of defined lncRNA sequence elements essential for site specific targeting of SUV39H1 to the *Oct4/OCT4* promoter.

**A deletion analysis identifies *mOct4P4* lncRNA regions essential for Oct4 silencing.** The MS2 RNA tagging system enabled us to demonstrate that a *mOct4P4* lncRNA–SUV39H1 complex locates to the promoter of the ancestral *Oct4* gene in trans[17]. In order to identify lncRNA regions essential for *mOct4P4* function we used a mESC cell line stably expressing a flag-tagged version of the MS2 phage coat protein (MS2-flag mESCs) as well as *mOct4P4* deletion constructs that were tagged with 24 repeats of the MS2 RNA stem loop motif (Fig. 2a, Supplementary Fig. 2a). To ensure nuclear localization, ectopically expressed lncRNAs contained *mOct4P4* regions corresponding to the 5' and 3' UTR regions of parental *Oct4*, previously shown to determine nuclear restriction of the endogenous *mOct4P4* lncRNA (Fig. 2a)[17].

Established stable mESC cell lines displayed nuclear enrichment of all *mOct4P4*-24xMS2 lncRNA versions (Fig. 2b, c). Ectopic expression of full length *mOct4P4*-24xMS2 but also deletion constructs with 200 or 400 nucleotides deletions located at the *mOct4P4* 3' terminus (Δ200, Δ400) efficiently reduced *Oct4* mRNA levels, as determined by quantitative RT-PCR (Fig. 2d). Remarkably, constructs lacking 600 nucleotides or more extended 3' lncRNA regions were no longer able to reduce *Oct4* RNA expression (Δ600, Δ800, Δ994; 5′ + 3′; Fig. 2d). These results were recapitulated by western blotting using an OCT4 specific antibody (Fig. 2e). These data indicate that functionally relevant *mOct4P4* sequence elements are anticipated to be located between position 984 and 1188 of the mature *mOct4P4* lncRNA.

We next evaluated the ability of *mOct4P4*-24xMS2 deletion construct derived lncRNAs to (i) tether the flag-tagged MS2 phage coat protein to the *Oct4* promoter and (ii) trigger increased H3K9me3 levels at the *Oct4* promoter. Anti-flag ChIP experiments revealed that only MS2 RNA tagged full length, Δ200 and Δ400 *Oct4P4*-24xMS2 lncRNAs were able to locate the flag-tagged MS2 protein to the promoter of the ancestral *Oct4* gene

and to trigger a local increase of H3K9me3 (Fig. 2f, g, Supplementary Fig. 2b). Accordingly, MS2 RNA tagged *Oct4P4* lncRNA versions that failed to suppress *Oct4* expression (Δ600, Δ800, Δ994; 5′ + 3′; Fig. 2d, e) were unable to locate flag-tagged MS2 and H3K9me3 to the *Oct4* promoter (Fig. 2f, g). Of notice, ectopically expressed full length *mOct4P4*-24xMS2 lncRNA was exclusively recruited to the *Oct4* promoter but not to the promoters of *Daxx*, *H2Q10*, *Ceher1*, *Pp1r18*, and *Rab5A* genes that are localized up- and downstream of *Oct4* on chromosome 17 (Supplementary Fig. 2c).

Together, this indicates that a 200 nucleotide sequence spanning position 984–1183 of the *mOct4P4* lncRNA has a central role in orchestrating target site specific epigenetic silencing of the ancestral *Oct4* gene in trans.

**A 200 nucleotide *mOct4P4* region is sufficient to silence parental gene expression.** To directly test the importance of the 200 nucleotide *mOct4P4* lncRNA region, we generated expression constructs encoding MS2 RNA-stem loop tagged m*Oct4P4* that lacks the relevant 200 nucleotide region (−200 bp -*mOct4P4*-24xMS2) but also a construct encoding a MS2 RNA motif tagged *mOct4P4* lncRNA version that exclusively covers region 984–1183 (200 bp-*mOct4P4*-24xMS2) (Supplementary Fig. 2d). Both constructs contained the 5' and 3' *mOct4P4* regions to ensure nuclear localization of expressed lncRNAs (Fig. 3a). Stable MS2-flag mESC clones expressed −200 bp-*Oct4P4*-24xMS2, 200 bp-*Oct4P4*-24xMS2, or full length *mOct4P4*-24xMS2 lncRNAs as nuclear restricted RNAs, as validated by quantitative RT-PCR (Fig. 3a–c).

We next tested the impact of ectopic lncRNA expression on H3K9me3 mediated silencing of parental *Oct4*. Quantitative RT-PCR and western blotting revealed that −200 bp-*mOct4P4*-24xMS2 expression failed to silence *Oct4* expression. In contrast, expression of the 200bp-*mOct4P4*-24xMS2 lncRNA version mediated efficient suppression of ancestral *Oct4*, recapitulating silencing by full length *mOct4P4*-24xMS2 lncRNA (Fig. 3d, e). In line with this, ChIP experiments showed that the 200bp-*mOct4P4*-24xMS2 lncRNA, but not the -200bp-*mOct4P4*-24xMS2 - version recruited MS2-flag and H3K9me3 to the parental *Oct4* promoter (Fig. 3f, g).

We conclude that *mOct4P4* pseudogene lncRNA contains two regions with an essential role in silencing of the ancestral *Oct4* gene: (i) 5' and 3' located sequences to ensure nuclear lncRNA and (ii) region 984–1183 that directs H3K9me3 to the *Oct4* promoter.

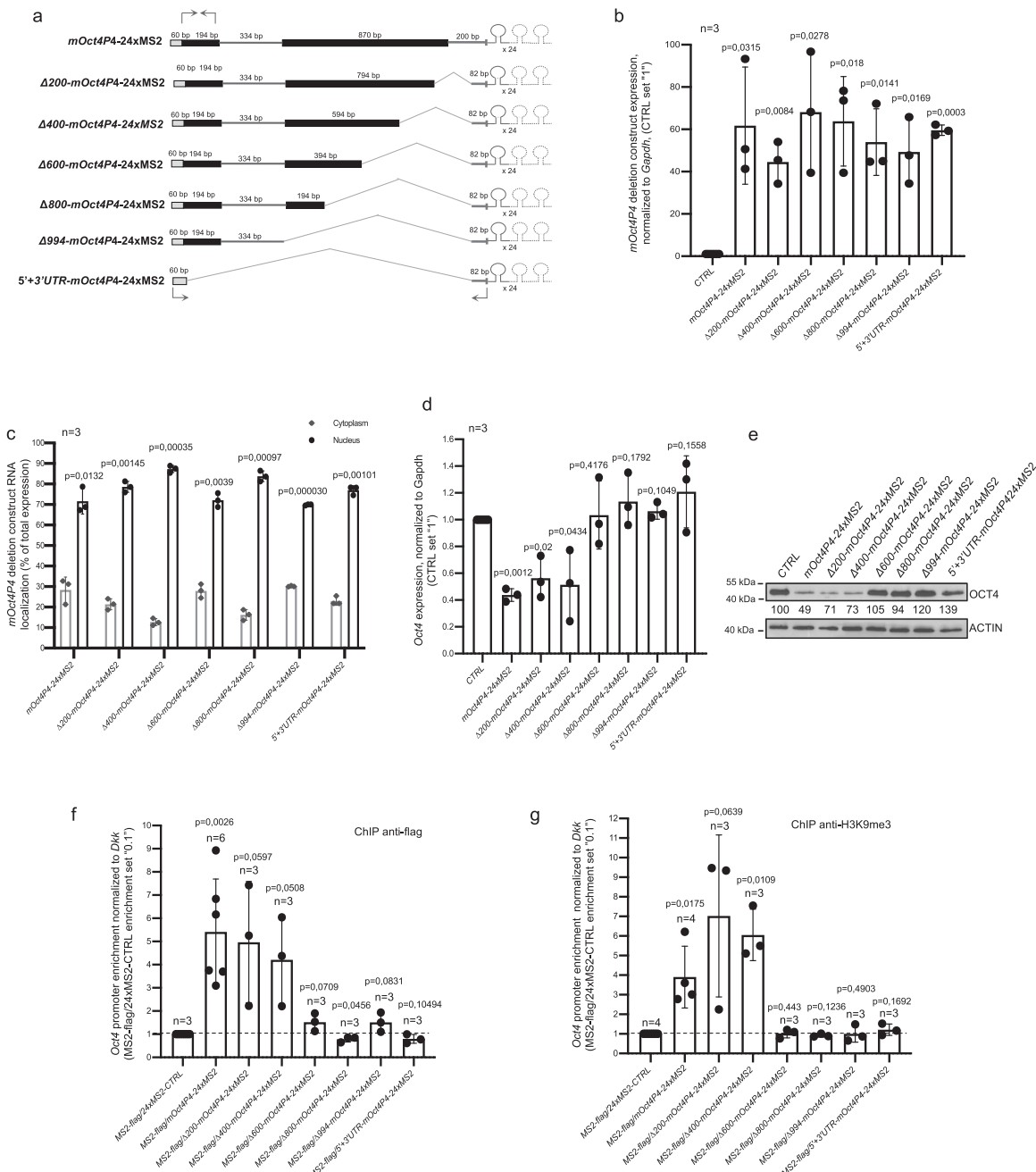

**Fig. 2 mOct4P4 deletion analysis identifies the minimal RNA regions essential for lncRNA function. a** Schematic representation of the *mOct4P4*-24xMS2 deletion constructs. Gray boxes, sequences with homology to *Oct4/OCT4* 5'UTR; gray lines, sequences with homology to *Oct4/OCT4* 3'UTR; 334 bp lines represent centrally located, spliced fragment present in *mOct4P4*. 24xMS2 RNA stem loop motifs located at the 3' end of *mOct4P4* deletion constructs are indicated. Arrows indicate the locations of RT-PCR primers used to amplify *mOct4P4* deletion constructs. **b** Levels of ectopic expression of *mOct4P4*-24xMS2 deletion construct (**a**) in mESCs, as determined by qRT-PCR. *p* Values relate to CTRL. **c** Subcellular localization of lncRNAs derived from *mOct4P4*-24xMS2 lncRNA deletion constructs (**a**) in mESCs, as determined by qRT-PCR. Expression values are shown as percentage of total RNA levels. *p* Values indicate significant nuclear versus cytoplasmic localization. **d** *Oct4* mRNA levels in mESCs with ectopic expression of *mOct4P4* deletion constructs (**a**), as determined by qRT-PCR. Expression values were normalized against gapdh. *p* Values relate to CTRL. **e** Representative image of OCT4 western blotting analysis in mESC cells overexpressing *mOct4P4* deletion constructs shown in (**a**). ACTIN was used as loading control. Numbers represents values of OCT4 expression as mean of three independent experiments (control was set "100"). **f, g** anti-flag ChIP (**f**) and anti H3K9me3 ChIP (**g**) on the *Oct4* promoter region in mESCs stably overexpressing 24xMS2 tagged deletion constructs shown in (**a**). qRT-PCR was performed to measure promoter enrichment; *p* values relate to MS2-flag/24xMS2-CTRL. Precise *p* values are indicated; *n* number of independent experiments carried out.

**FUS interacts with endogenous *mOct4P4* to allow parental *Oct4* gene silencing**. In order to obtain additional insights into the mechanism of *mOct4P4* lncRNA mediated silencing of *Oct4* we aimed to identify *mOct4P4* lncRNA interacting proteins. MS2-flag cells expressing full-length *mOct4P4*-24xMS2 and control mESCs expressing only a 24xMS2 stem loop control RNA were used to perform anti-flag RNA immunoprecipitation (RIP) experiments. Obtained control and *mOct4P4*-24xMS2 RNA-immunoprecipitates where run on denaturing polyacrylamide gels. After Coomassie staining, protein bands

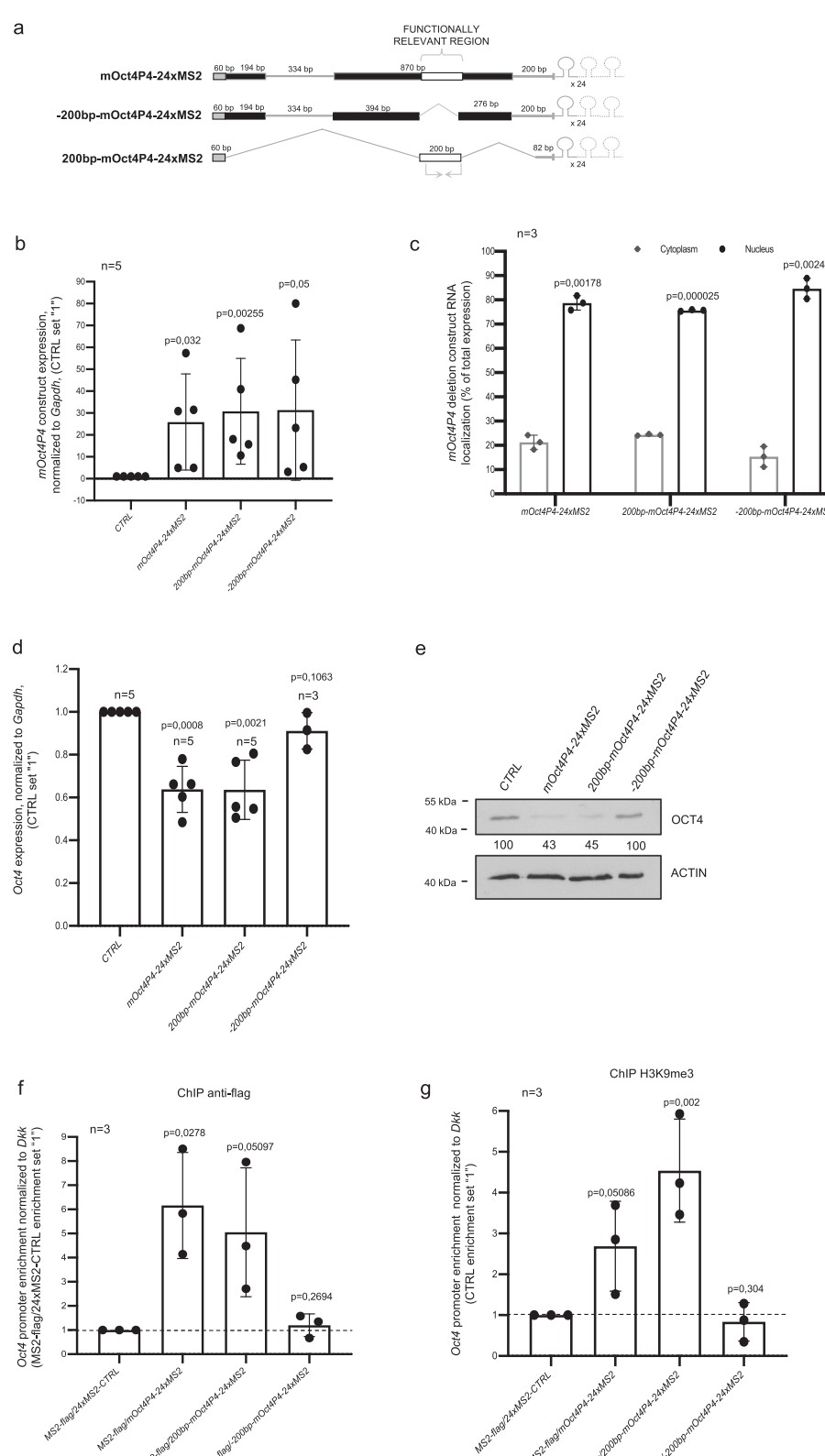

specifically appearing in eluates from *mOct4P4*-24MS2 RIPs were cut out from the gel and subjected to mass spectrometry (Fig. 4a). Flag-tagged MS2 as well as an additional set of proteins were shown to be specifically over-represented in analyzed protein bands obtained from *mOct4P4* lncRNA RIP eluates (Fig. 4a, Supplementary Table 1a, Supplementary Data 1). Given the reported involvement in gene silencing, we

focused our interest on the RNA and DNA binding protein FUS[28,29]. In addition to transcriptional regulation, FUS has been demonstrated to be involved in DNA repair, alternative splicing, transcriptional regulation, RNA localization and stress granules[30]. FUS translocation events and mutations have been linked with liposarcoma and amyotrophic lateral sclerosis, respectively[31–33].

**Fig. 3 Silencing and promoter targeting is limited to a 200 nucleotide *mOct4P4* lncRNA region. a** Schematic representation of generated *mOct4P4*-24xMS2 deletion constructs. Gray boxes, sequences with homology to *Oct4/OCT4* 5′UTR; gray lines, sequences with homology to *Oct4/OCT4* 3′UTR. 334 bp-spliced sequences are present in *mOct4P4* and −200*mOct4P4* sequences; deleted regions are indicated. White boxes indicate identified 200 nucleotide *mOct4P4* region. 24xMS2 RNA stem loop motifs at the 3′ end of *mOct4P4* deletion constructs are indicated. Arrows indicate the locations of RT-PCR primers. **b** qRT-PCR determining *200 bp-mOct4P4* and *−200 bp-mOct4P4* expression levels in experimental mESCs. Expression values were normalized to *Gapdh*. **c** Subcellular localization of lncRNAs derived from constructs in (**a**). Expression values are shown as percentage of total RNA levels, as determined by qRT-PCR. **d**, **e** qRT-PCR (**d**) and western blot analysis (**e**) using mESCs ectopically expressing *mOct4P4*, *200 bp-mOct4P4*, and *−200 bp-mOct4P4* constructs. *Oct4* expression values were normalized against *Gapdh* (**d**) or *ACTIN* (**e**). Shown numbers represent *OCT4/ACTIN* ratio as mean of three independent experiments (control was set "100") (**e**). **f**, **g** ChIP analysis of *Oct4* promoter region in mESCs stably overexpressing indicated constructs and using described antibodies. qRT-PCR was performed to measure promoter enrichment. Only *mOct4P4* and *200 bp-mOct4P4* constructs localize to the *Oct4* promoter (**f**) and drive H3K9me3 enrichment (**g**). Error bars represent standard deviation. Precise *p* values are indicated. n: number of independent experiments carried out.

Validation of RIP eluates by western blotting and RT-PCR confirmed interaction of FUS with the full length *mOct4P4* lncRNA (Fig. 4b). We were also able to detect *mOct4P4*-24xMS2 lncRNA as well as MS2-flag protein in the eluates from anti-FUS RIP experiments, corroborating FUS–*Oct4* pseudogene lncRNA interaction (Fig. 4c).

We were next interested in evaluating whether FUS is required for *mOct4P4* lncRNA mediated silencing of *Oct4*. Transient knockdown of *FUS* abolished *mOct4P4* function, thus rescuing OCT4 protein expression in *mOct4P4*-24xMS2 lncRNA over-expressing mESCs (Fig. 4d). In line with this, ChIP experiments revealed that FUS localizes to the *Oct4* promoter in MS2-flag mESCs overexpressing *mOct4P4*-24xMS2 (Fig. 4e).

We previously showed that the *mOct4P4* lncRNA is essential to maintain SUV39H1-dependent silencing of parental *Oct4* in primary mouse embryonic fibroblasts (pMEFs), indicating that persistent localization of the *mOct4P4* lncRNA at the *Oct4* promoter is essential to maintain *Oct4* silencing in differentiated cells[17].

To test whether *mOct4P4* lncRNA is required for the localization of FUS to the *Oct4* promoter we performed ChIP experiments in *mOct4P4* lncRNA knock-down pMEFs. Our results show that loss of endogenous *mOct4P4* lncRNA displaced FUS from the *Oct4* promoter in pMEFs (Fig. 4f). Accordingly, siRNA mediated depletion of *FUS* from pMEFs significantly increased *Oct4* mRNA expression, recapitulating the effect of *mOct4P4* knockdown on parental gene expression (Fig. 4g, h). This effect was paralleled by increased expression of self-renewal transcription factors *Sox2*, *Nanog* and *Klf4* (Fig. 4i). We conclude that FUS is essential for the initiation and maintenance of *mOct4P4* lncRNA mediated silencing of *Oct4* in order to suppress self-renewal circuits in differentiated mouse cells.

**FUS facilitates binding of SUV39H1 to the *mOct4P4* lncRNA**. *mOct4P4* deletion constructs revealed that crucial regions for *Oct4P4* function are limited to a 200 nucleotide region, spanning positions 984–1183 (Fig. 3a, d–g). To test whether this RNA region interacts with SUV39H1 or FUS, we performed anti-SUV39H1 and anti-FUS RIP experiments using MS2-flag mESC clones overexpressing full length *mOct4P4*-24xMS2, *−200 bp-mOct4P4*-24xMS2 or *200 bp-mOct4P4*-24xMS2 constructs.

We found that the SUV39H1 protein co-immunoprecipitated with the full-length *mOct4P4*-24xMS2 and *200 bp-mOct4P4*-24xMS2 lncRNAs, but not with *−200 bp-mOct4P4*-24xMS2 lncRNA (Fig. 5a). Interestingly, all types of ectopically expressed *mOct4P4* lncRNAs versions bound FUS in RIP experiments, suggesting that FUS binds multiple *mOct4P4* lncRNA regions (Fig. 5b). In contrast, *mOct4P4*–SUV39H1 interaction critically depends on the presence of the 200 nucleotide motif. Notably, we did not find evidence for direct interaction of SUV39H1 and FUS in co-immunoprecipitation assays (Supplementary Fig. 3a, b).

In addition, we did not find SUV39H1 peptides in our mass spectrometry data from *mOct4P4*-24xMS2 lncRNA pull down experiments (Supplementary Data 1). This is in line with a lack of SUV39H1 in published data on the FUS interacting proteome[34–38]. We conclude that direct SUV39H1–FUS interaction is not a pre-requisite for silencing complex formation.

To study the functional interplay between FUS, SUV39H1, and *mOct4P4* in determining silencing complex function, we first performed anti-FUS RIP in *Suv39h1* knockdown MS2-flag mESCs, stably overexpressing full-length *mOct4P4*-24xMS2 or *200 bp-mOct4P4*-24xMS2 lncRNAs. We found that FUS interacts with the full length *mOct4P4*-24xMS2 but also *the 200 bp-mOct4P4* lncRNA version in the presence and absence of SUV39H1 (Fig. 5c, Supplementary Fig. 3c). Thus, SUV39H1 is dispensable for FUS–*mOct4P4* lncRNA interaction.

In a second step we transiently depleted FUS from experimental cells and performed anti-SUV39H1 RIP experiments followed by *mOct4P4* specific RT-PCR. We found that loss of FUS abolishes SUV39H1 binding to the full length *mOct4P4* lncRNA (Fig. 5d, Supplementary Fig. 3d). Strikingly, binding of SUV39H1 to the *200 bp-mOct4P4*-MS2 lncRNA (*mOct4P4* positions 984–1183) does not require FUS (Fig. 5d). This indicates that binding of FUS to the full-length *mOct4P4* lncRNA plays an important role in providing access for SUV39H1 to the 200 nucleotide region. However, in the context of reduced lncRNA sequence complexity of the *200 bp-mOct4P4*-MS2 construct, the critical 200 nucleotide region appears to be directly accessible to SUV39H1, rendering the action of FUS dispensable.

*Oct4* mRNA and *mOct4P4* lncRNA share high sequence identity levels, raising the question as to whether SUV39H1 and FUS may also interact with the endogenous *Oct4* mRNA.

Importantly, RIP experiments using mESCs demonstrated that under our experimental conditions SUV39H1 and FUS display binding specificity towards *mOct4P4* lncRNA but not *Oct4* or other mRNAs such as *Sox2*, *Nanog*, *Gapdh*, or *Actin* (Fig. 5e, f).

This demonstrates that sequence degeneration after *mOct4P4* pseudogene formation resulted in the formation of binding sites for FUS and SUV39H1, conferring a new biological function to the *mOct4P4* lncRNA. On the mechanistic level, our data indicate that FUS has a critical role in supporting the interaction of SUV39H1 with full length *mOct4P4* lncRNA, suggesting that FUS licenses the formation of a functional SUV39H1–*mOct4P4* lncRNA complex in mESCs.

**FUS mediates targeting of SUV39H1 by *mOct4P4* lncRNA to the *Oct4* promoter**. We next wished to investigate how lncRNA: protein binding requirements translate into site specific targeting of a SUV39H1 containing silencing complex to the *Oct4* promoter. We first validated whether FUS has a role in directing *mOct4P4* lncRNA and SUV39H1 to the *Oct4* promoter.

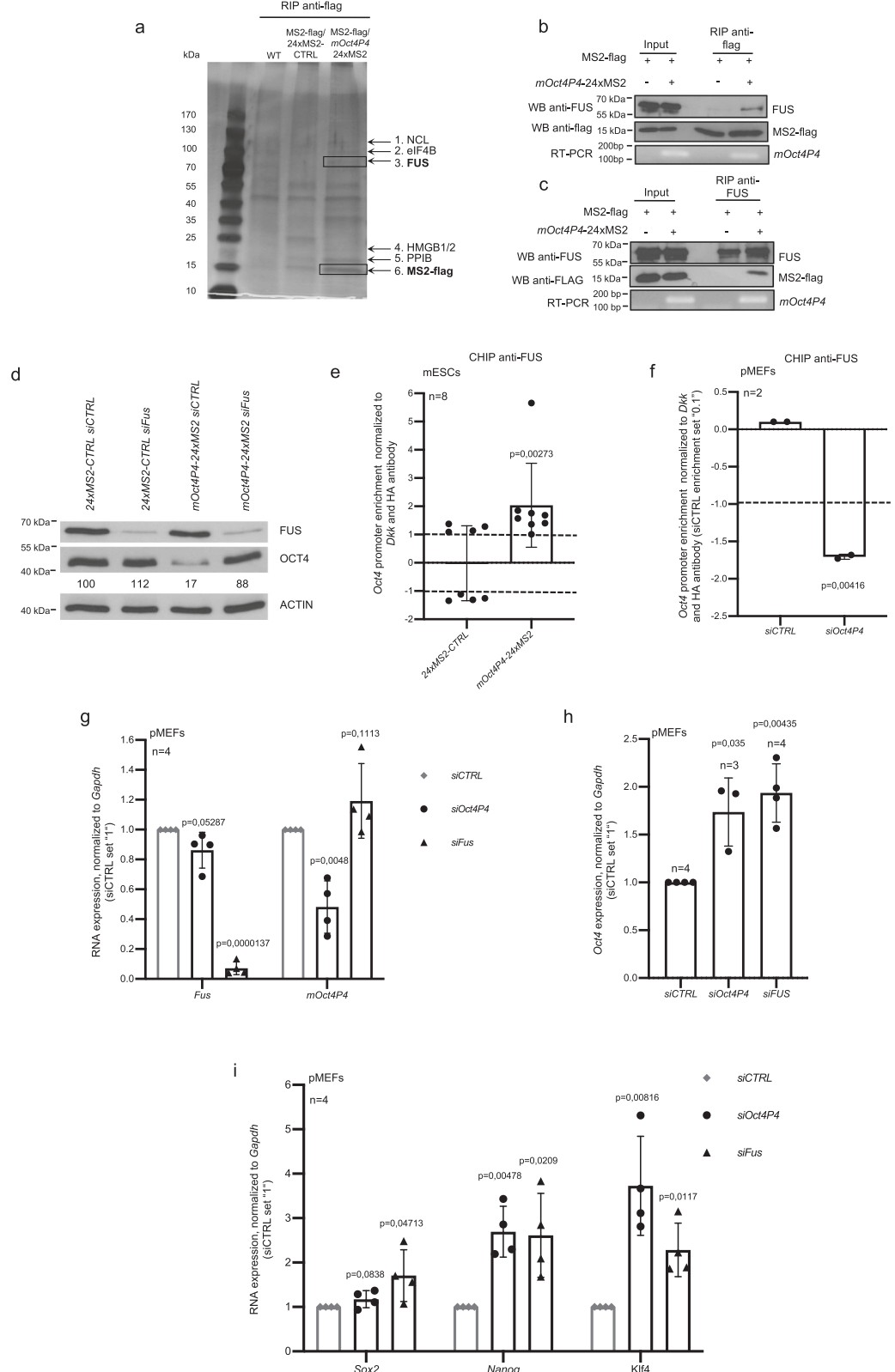

Anti-flag ChIP experiments using chromatin from flag-MS2 mESCs expressing MS2-RNA tagged *mOct4P4* lncRNA variants demonstrate that transient knockdown of *Fus* abolishes the recruitment of full length *mOct4P4*-24xMS2 but also SUV39H1 to the promoter of the ancestral *Oct4* gene (Fig. 6a, b). In line with this, ChIP revealed reduced H3K9me3 levels at the *Oct4* promoter and increased OCT4 protein expression

after siRNA mediated depletion of *Fus* from full length *mOct4P4* lncRNA overexpressing mESCs (Fig. 6c). This demonstrates that FUS is essential for targeting the endogenous *mOct4P4* –lncRNA–SUV39H1 complex to the Oct4 promoter.

Importantly, performing anti-flag ChIP we found that siRNA mediated depletion of *Fus* does not impair the localization of the 200 bp-*mOct4P4*-24xMS2 lncRNA version to the Oct4 promoter

**Fig. 4 FUS is required for mOct4P4 lncRNA-mediated silencing of Oct4 in mESCs. a** Silver stained protein gel of eluates obtained from *mOct4P4-24xMS2* anti-flag RIP experiments. mESCs expressing flag-MS2 and *mOct4P4-24xMS2* were used. Indicated bands specifically elute from *mOct4P4-24xMS2* lncRNA. Protein identity was determined by mass spectrometry (Supplementary methods). **b** Anti-flag RIP using mESCs expressing MS2-flag/full length *mOct4P4-24xMS2* or *24xMS2* RNA control cells using anti-flag antibody. Agarose gel electrophoresis after quantitative RT-PCR demonstrates the presence of *mOct4P4-24xMS2* stem loop RNA (bottom panel). Detection of FUS and MS2-flag proteins by Western blotting (top and middle panel respectively). Bands analyzed by mass spectrometry are indicated as numbers (1–6); complete data on protein identification is available in the provided Supplementary Data 1. **c** Anti-FUS RIP using MS2-flag mESCs expressing full length *mOct4P4-24xMS2* or *24xMS2* RNA control. Presence of FUS and flag-MS2 in eluates was validated by western blotting (top and middle panel respectively). Quantitative RT-PCR followed by agarose gel electrophoresis verified the presence of *mOct4P4-24xMS2* in anti-FUS RIP experiments (bottom panel). **d** FUS and OCT4 western blotting using eluates from *mOct4P4-24xMS2* or *24xMS2* mESCs transiently transfected with the indicated siRNAs. ACTIN was used as loading control. Numbers represent OCT4/ACTIN ratio as mean of three independent experiments (24xMS2-CTRL siCTRL was set "100"). **e, f** ChIP analysis of *Oct4* promoter region using an anti-FUS antibody in control or FUS knockdown mESCs (**e**) or pMEFs (**f**). Eluates were analyzed by qRT-PCR. **g** *Fus* and *mOct4P4* expression levels in pMEFs transiently transfected with indicated siRNAs, as determined by qRT-PCR. Expression levels were normalized to *Gapdh*. **h, i** qRT- PCR analysis using pMEF cells subjected to siRNA-mediated knockdown of *mOct4P4* and *Fus*. Expression values for *Oct4* (**h**) or self-renewal markers (**i**) were normalized against *Gapdh*. Error bars represent standard deviation. Precise *p* values are indicated. *n* number of independent experiments carried out.

of experimental mESCs (Fig. 6d). Accordingly, *200 bp-mOct4P4* overexpression results H3K9me3 enrichment at the *Oct4* promoter and a reduction of OCT4 protein expression in control but also *Fus* knockdown mESCs (Fig. 6e, f). Thus, FUS is dispensable for parental *Oct4* silencing in the context of the minimal sufficient 200 nucleotide *mOct4P4* construct. However, in context of the increased sequence complexity of endogenous, full-length *mOct4P4*, FUS is essential to license the interaction between SUV39H1 and *mOct4P4* to allow the formation of a silencing complex with *Oct4* promoter target-specificity.

To further dissect requirements for *Oct4* promoter targeting we evaluated the relevance of SUV39H1 for targeting FUS and *mOct4P4* lncRNA to the parental *Oct4* gene. Anti-FUS ChIP experiments revealed that siRNA mediated depletion of *Suv39h1* delocalizes FUS from the *Oct4* promoter in mESCs ectopically expressing full length *mOct4P4* or the *200 bp-Oct4P4* lncRNA (Fig. 6g). Importantly, siRNA mediated knockdown of *Suv39h1* abrogates the localization of full length *mOct4P4-24xMS2* but also *200 bp-mOct4P4-24xMS2* lncRNA versions to the promoter of the ancestral *Oct4* gene, as demonstrated by anti-flag ChIP. This effect was linked with impaired imposition of H3K9me3 to the *Oct4* promoter and loss of parental *Oct4* silencing in both experimental cell lines (Fig. 6h–k, Supplementary Fig. 4).

These data highlight that FUS is essential to instruct the loading of the repressive SUV39H1 HMTase to the critical 200 *mOct4P4* lncRNA nucleotide region. This FUS dependent step is central to program target specificity of SUV39H1, towards the promoter of the parental *Oct4* gene.

**Functional conservation of a FUS–SUV39H1–OCT4 pseudogene lncRNA silencing complex.** After identifying critical players for *mOct4P4* function we set out to test whether all critical mechanistic steps are conserved in human OVCAR-3 cells. We first generated OVCAR-3 cell lines stably transfected with an expression vector encoding 24xMS2 tagged full-length *hOCT4P3* (*hOCT4P3-24xMS2*) or a 24xMS2 tagged *hOCT4P3* lncRNA region (*200 bp-hOCT4P3-24xMS2*) that corresponds to the functional relevant 200 nucleotide *mOct4P4* region (Fig. 7a, Supplementary Fig. 5a). Functional experiments were carried out after transiently transfecting experimental cell lines with an expression vector encoding flag-tagged MS2.

In line with data from OVCAR-3 cells overexpressing untagged *hOCT4P3* (Fig. 1c), we found that ectopic *hOCT4P3-24xMS2* expression reduced the expression of endogenous *OCT4/OCT4* on the RNA and protein level (Fig. 7b, c). Anti-flag RIP revealed interaction of MS2-flag with ectopically expressed *hOCT4P3-24xMS2* lncRNA, as demonstrated by RT-PCR (Fig. 7d).

ChIP experiments using anti-flag and anti-H3K9me3 specific antibodies showed that the *hOCT4P3-24xMS2* lncRNA localizes the flag-tagged MS2-protein to the promoter of the ancestral *OCT4* gene, triggering a local increase in H3K9me3 (Fig. 7e, f). In line with this, western blotting and RT-PCR on protein and RNA fractions from anti-flag RIP eluates revealed that SUV39H1 and FUS co-immunoprecipitate with full length *hOCT4P3-24xMS2* lncRNA (Fig. 7g, h).

Anti-SUV39H1 RIP experiments in control MS2-flag and MS2-flag/*hOCT4P3-24xMS2* OVCAR-3 cells demonstrated that siRNA-mediated depletion of FUS disrupts binding of SUV39H1 to the full-length *hOCT4P3* lncRNA (Fig. 7i, Supplementary Fig. 5b). In line with data from mESCs, transient depletion of *FUS* disrupts pseudogene lncRNA mediated reduction of OCT4 expression in full-length *hOCT4P3-24xMS2* lncRNA overexpressing OVCAR-3 cells (Fig. 7j).

To validate the selective requirement of FUS for licencing full length *OCT4* pseudogene lncRNA function in human cells we generated OVCAR-3 cells stably expressing a 24xMS2 tagged, 200 nucleotide *hOCT4P3* region corresponding to the respective sequence stretch in the *mOct4P4* lncRNA (Fig. 7a, Supplementary Fig. 5c). Importantly, we found that ectopic *200 bp-hOCT4P3* lncRNA expression recapitulates *OCT4* silencing triggered by full length *hOCT4P3* lncRNA (Fig. 7k). In line with data from mESCs, *OCT4* silencing triggered by the *200 bp-hOCT4P3-24xMS2* lncRNA version was independent of FUS expression in OVACR-3 cells (Fig. 7l, Supplementary Fig. 5d, e).

We conclude that all aspects of *mOct4P4* function are recapitulated by *hOCT4P3* in human cells. This demonstrates that pseudogene lncRNA dependent silencing of *Oct4/OCT4* represents an evolutionary conserved mechanism to fine-tune the expression of the parental *Oct4/OCT4* gene.

On the mechanistic level we propose a model where FUS binding to the endogenous *mOct4P4/hOCT4P3* lncRNA plays an important role in rendering the 200-nucleotide region accessible for SUV39H1 binding. This step is essential to license the formation of a SUV39H1 HMTase containing silencing complex with programmed target specificity towards the parental *Oct4/OCT4* promoter (Fig. 8).

## Discussion

Here, we investigate the molecular mechanism and evolutionary conservation of *Oct4/OCT4* pseudogene lncRNA mediated control of parental gene expression. Repression of *hOCT4P3* or *mOct4P4* lncRNA expression in human OVCAR-3 or mESCs using the CRISPR/dCas9-HAKRAB system resulted in loss of H3K9me3 at the *OCT4/Oct4* promoter and elevated OCT4/Oct4 expression levels (both at RNA and protein levels) in human or

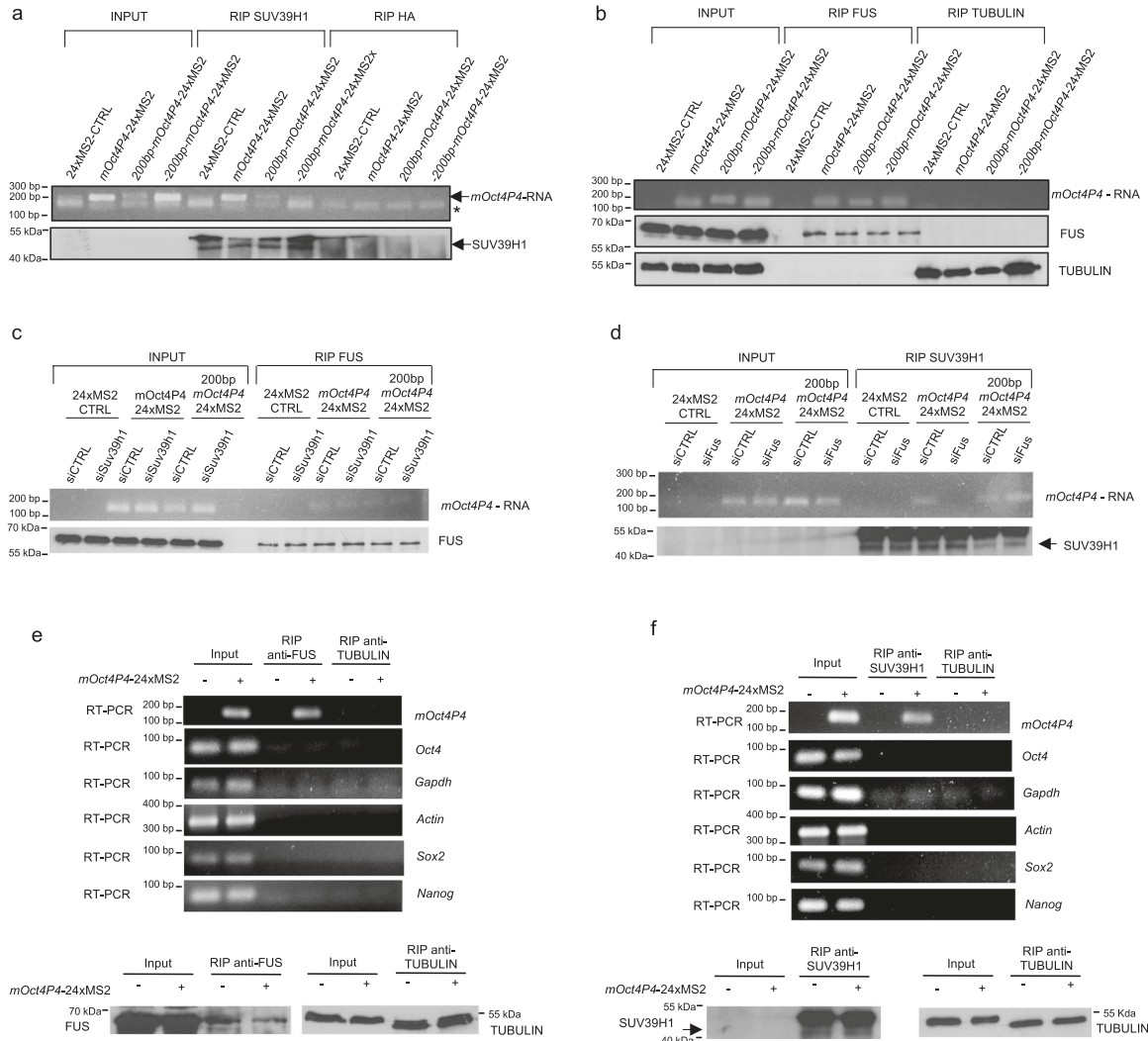

**Fig. 5 FUS licenses the formation of a SUV39H1–*mOct4P4* lncRNA silencing complex. a** Anti-SUV39H1 RIP in mESCs stably expressing *mOct4P4*-24xMS2, *200 bp-mOct4P4*-24xMS2 or −*200 bp-mOct4P4*-24xMS2 lncRNAs. Agarose gel electrophoresis after *mOct4P4* specific quantitative RT-PCR (top panel) and anti-SUV39H1 western blotting (bottom panel) on RIP eluates are shown. Anti-HA RIP was used as negative control. Arrow indicates primer dimers. **b** Anti-FUS RIP in mESCs stably expressing *200 bp-mOct4P4*-24xMS2 or −*200 bp-mOct4P4*-24xMS2 lncRNAs. Agarose gel electrophoresis after *mOct4P4* specific quantitative RT-PCR (top panel) and anti-FUS western blotting (bottom panel) on RIP eluates are shown. Anti-TUBULIN RIP was used as negative control. **c** Anti-FUS RIP in mESCs stably expressing *mOct4P4*-24xMS2 or *200 bp-mOct4P4*-24xMS2 lncRNAs. Agarose gel electrophoresis after qRT-PCR confirmed FUS-*Oct4P4* binding (top panel). Immunoprecipitation of FUS was validated by western blotting (bottom panel). **d** Anti-SUV39H1 RIP in Fus knockdown *mOct4P4*-24xMS2 and *200 bp-mOct4P4*-24xMS2 mESCs. *mOct4P4* specific qRT-PCR followed by agarose gel electrophoresis is shown (top panel). Presence of SUV39H1 in RIP eluates was validated by Western blotting (bottom panel). **e, f** Anti-SUV39H1 RIP (e) and anti-FUS RIP (f) in mESCs stably expressing *mOct4P4*-24xMS2 or 24xMS2. Agarose gel electrophoresis after *Oct4* mRNA or *mOCT4P4* specific qRT-PCR is shown (top panels). Anti-TUBULIN RIP was used as negative control. Immunoprecipitation of SUV39H1 (**e**) or FUS (**f**) was validated by Western blotting (bottom panels **e** and **f**).

mouse cells, respectively. This indicates functional conservation of *Oct4* pseudogene lncRNA mediated silencing of parental gene expression in mouse and human cells. High overall sequence identity and conservation of *mOct4P4* function in human cells suggested the existence of functionally relevant lncRNA regions.

A deletion analysis identified a 200-nucleotide region in *mOct4P4* and *hOCT4P3* lncRNA that is required for targeting of the lncRNA-SUV39H1 silencing complex to the promoter of the ancestral *Oct4/OCT4* gene, resulting in local H3K9 tri-methylation. Binding of *Oct4/OCT4* pseudogene lncRNA by SUV39H1 is in line with studies demonstrating interaction of SUV39H1 HMTases with pericentric RNAs, telomere repeat containing RNA (*TERRA*), *LINE1 L1MdA* 5′UTR elements, *SINE B1* repeats and pRNAs of the rRNA cluster[39–42]. Direct

interaction of *mOct4P4* lncRNA with SUV39H1 was recently demonstrated by in vitro EMSA experiments (37). SUV39H1 HMTase–RNA-binding specificity is reported to be promiscuous and characterized by low sequence specificity. This lead to the hypothesis that the formation of lncRNA–SUV39H HMTase complexes with defined epigenetic function may depend on additional proteins or the presence of physiologically functional RNA:chromatin templates[43,44].

RNA pull-down experiments revealed a series of *mOct4P4* lncRNA interacting proteins with a potential role in silencing parental *Oct4*. Here, we demonstrate that the RNA binding protein FUS has a critical role in *Oct4/OCT4* lncRNA mediated silencing of OCT4. Loss of FUS prevents the formation of a full length *mOct4P4/hOCT4P3* lncRNA–SUV39H1 silencing

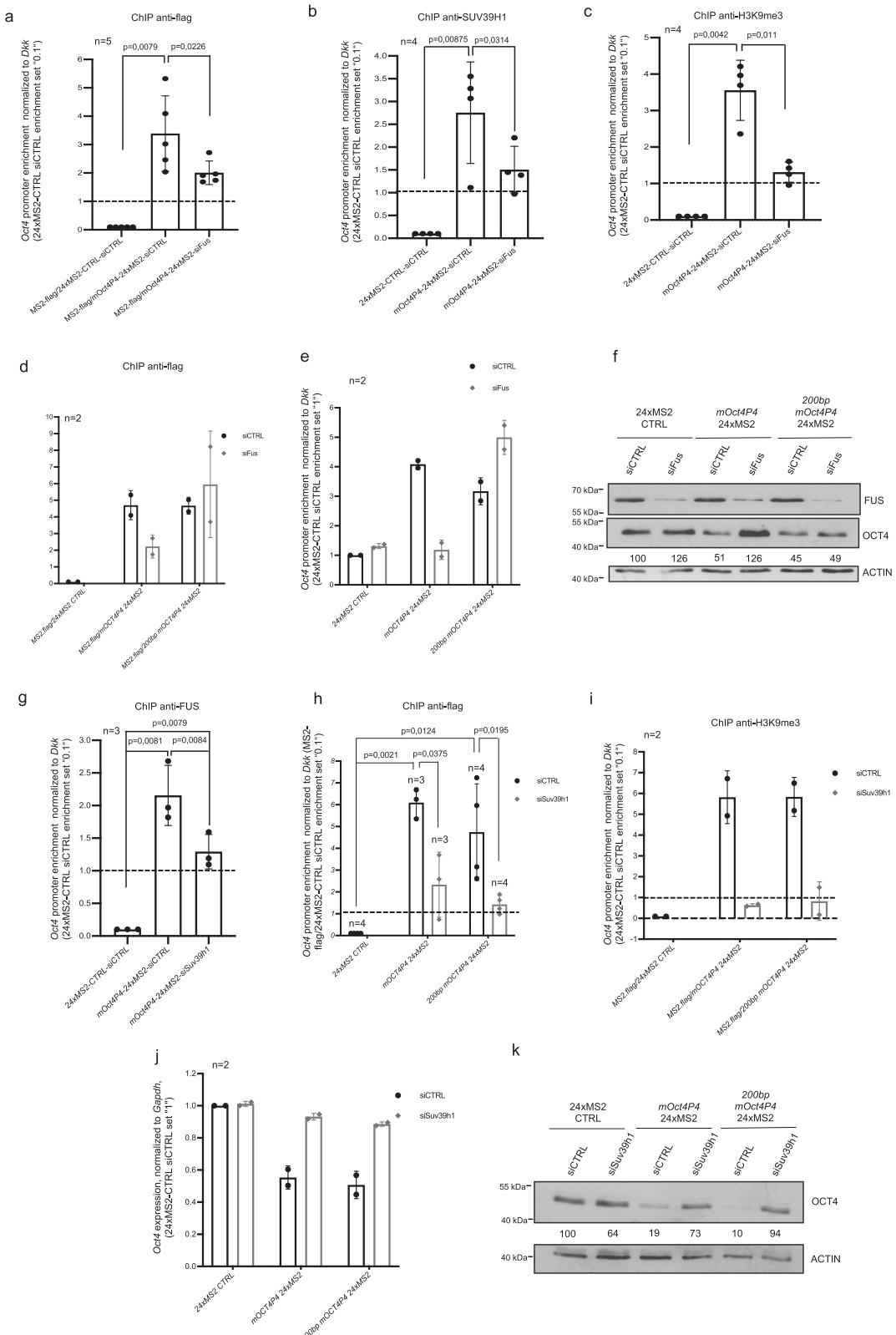

complex, abrogating the initiation and maintenance of *Oct4*/*OCT4* silencing. Notable, FUS is dispensable for the function of the minimal sufficient *mOct4P4*/*hOCT4P3* lncRNA version (*200 bp-mOct4P4; 200 bp-hOCT4P3*). Thus, we conclude that FUS does not have a central role in closing the *Oct4*/*OCT4* promoter.

We propose that FUS is critical for the structuring the long *Oct4* pseudogene lncRNA template to allow the binding of SUV39H1 to the 200-nucleotide region, thereby defining a specialized SUV39H1–lncRNA complex with selective target specificity towards the parental *Oct4*/*OCT4* promoter. Importantly, FUS and SUV39H1 do not bind to the *Oct4* mRNA in RIP

**Fig. 6 SUV39H1 and the 200 bp-*mOct4P4* region are sufficient for paternal gene silencing. a–c** ChIP analysis of the *Oct4* promoter region in control, *mOct4P4*-24xMS2 or 24xMS2 overexpressing mESC lines, transfected with the indicated siRNAs. Antibodies used for RIP are shown. Quantitative RT-PCR was performed to evaluate enrichment of markers at the Oct4 promoter. **d, e** ChIP analysis of Oct4 promoter region in 24xMS2-CTRL, *mOct4P4*-24xMS2, and *200 bp-mOct4P4*-24xMS2 mESCs after siRNA mediated depletion of *Fus*. Quantitative RT-PCR was performed to measure abundance of flag-MS2 (**d**) and H3K9me3 (**e**) at the *Oct4* promoter. **f** OCT4 expression levels in 24xMS2-CTRL, *mOct4P4*-24xMS2 and 200 bp-mOct4P4-24xMS2 mESCs after Fus knockdown. ACTIN was used as loading control. Numbers represent OCT4/ACTIN ratio (24xMS2-CTRL siCTRL was set "100"). **g** ChIP analysis of FUS abundance at the *Oct4* promoter region in control, *mOct4P4*-24xMS2 or *Suv39h1* knockdown *mOct4P4*-24xMS2 overexpressing mESC lines. **h, i** ChIP analysis of *Oct4* promoter region in control, *mOct4P4*-24xMS2 and 200 bp-mOct4P4-24xMS2 mESCs after siRNA mediated depletion of *Suv39h1*. Quantitative RT-PCR was performed to measure enrichment of flag-MS2 (**h**) or H3K9me3 (**i**) at the *Oct4* promoter. **j, k** *Oct4* specific qRT-PCR (**j**) and OCT4 western blotting (**k**) of *Suv39h1* knockdown control, *mOct4P4*-24xMS2 and *200 bp-mOct4P4*-24xMS2 mESCs. RNA expression values were normalized against *Gapdh*; ACTIN was used as loading control in western blotting experiments. Numbers represents OCT4/ACTIN ratio (24xMS2-CTRL siCTRL was set "100"). Error bars represent standard deviation. Precise *p* values are indicated. *n* number of independent experiments carried out.

experiments. This demonstrates that the specific interaction with FUS and the noncoding RNA-guided SUV39H1 HMTase represents a new biological feature of *Oct4P4/OCT4P3* lncRNAs, that was acquired during pseudogene evolution. Future experiments will have to validate whether FUS has a more general role in epigenetic gene regulation by controlling the association of lncRNAs with epigenetic writers. In addition, the impact of *Oct4/OCT4* promoter associated pseudogene transcripts on transcriptional initiation and *Oct4/OCT4* promoter evasion remains an interesting issue to be addressed.

In contrast to the selective requirement of FUS for full length pseudogene lncRNA function, we found that SUV39H1 is essential for targeting of both, the full-length and 200 nucleotide *mOct4P4/hOCT4P3* lncRNA versions to the *Oct4/OCT4* promoter. Thus, after FUS dependent silencing complex formation, SUV39H1 and the 200 nucleotide *mOct4P4/hOCT4P3* lncRNA regions hold the information for selective targeting and epigenetic silencing of the parental *Oct4/OCT4* gene promoter.

The requirement of FUS as critical factor to license endogenous *mOct4P4/hOCT4P3* lncRNA function may also represent a regulatory mechanism that restricts pseudogene-lncRNA mediated silencing to a defined biological context. Along these lines, PRMT1 dependent arginine methylation of FUS was recently shown to prevent the interaction with the *CCND1* gene promoter-associated noncoding RNA-D (pncRNA-D), thereby blocking the repression of the HAT activity of the CBP/p300 HAT complex[28,29]. Addressing post-translational modifications of FUS may identify windows of *mOct4P4/hOCT4P3* function in development and disease.

In addition to *mOct4P4/hOCT4P3* also other pseudogene derived lncRNAs, such as *DUXAP8* and *DUXAP10* have been shown to interact with epigenetic writers[12,45,46]. However, *DUXAP* lncRNAs rather act as general scaffold for epigenetic regulatory complexes that do not selectively target the parental *DUXA* gene. In contrast, pseudogene *PTENP1* antisense transcripts drive DNMT1 dependent silencing of the parental *PTEN* gene by paring with the 5′UTR of the nascent, sense *PTEN* RNA[9,11]. We experimentally validated that *Oct4* and *mOct4P4* are exclusively transcribed in sense orientation, thus excluding extended RNA:RNA interactions[17]. Thus, *mOct4P4* and *hOCT4P3* represent pseudogene sense lncRNAs that use a conserved mechanism to target and remodel the chromatin status of the parental gene promoter, located on a different chromosome.

Altogether, we propose a four-step model: (i) FUS binds *mOct4P4/hOCT4P3* to (ii) allow SUV39H1 binding to the 200 nucleotide region, followed by (iii) sequence specific targeting of the *Oct4/OCT4* promoter, resulting in (iv) increasing local H3K9me3 and HP1 levels and *Oct4/OCT4* silencing (Fig. 8). The specific binding of SUV39H1 to H3K9me3 is anticipated to contribute to the maintenance of local heterochromatin structure at the *Oct4/OCT4* promoter[40,41].

Silencing of *Oct4/OCT4* in trans may depend on complex long-range chromatin interaction of involved (pseudo)gene–loci, alternative DNA structures or the recruitment of additional factors. Elucidating mechanisms that functionally connect pseudogenes loci with ancestral genes will provide new insights into the power of pseudogenes encoded lncRNAs in fine-tuning the expression of ancestral genes in development and disease.

## Methods

**Cell culture**. Feeder independent mESCs were cultured on 0.2% gelatin-coated plates using mESC self-renewal medium composed by Dulbecco's modified Eagle's medium (DMEM) (Lonza) supplemented with 15% knockout serum replacement (Gibco), 1% nonessential amino acids (Gibco), 1 mM sodium pyruvate (Gibco), 0.1 mM β-mercaptoethanol, 1% penicillin/streptomycin (Lonza) and 1000 U/ml mouse leukemia inhibitory factor[47]. OVCAR-3 cells were obtained from ATCC and cultured in RPMI-1640 medium (Lonza) supplemented with 20% (v/v) fetal bovine serum (FBS) (Lonza), insulin (10 μg/ml; I9278,Sigma) and 1% (v/v) penicillin/streptomycin (Lonza). Primary mouse embryonic fibroblasts (pMEFs) were generated in house from 13.5 d.p.c. C57BL/6 mouse embryos. pMEFs were maintained in culture in DMEM (Lonza) supplemented with 10% (v/v) FBS (Lonza) and 1% (v/v) penicillin/streptomycin (Lonza). Cell lines were maintained as monolayers at 37 °C in a humidified 5% CO$_2$ atmosphere.

mESCs differentiation was obtained with a DMEM supplemented with 15% ES cell certified serum (Invitrogen), 1% non-essential amino acids (Gibco), 1 mM sodium pyruvate (Gibco), 1% L-glutamine (Invitrogen), 0.1 mM β-mercaptoethanol, and 1% penicillin/streptomycin (Invitrogen). EBs were generated by cultivating 300 cells in hanging drops culture for 3 days. Subsequently, EBs were transferred to a low-attachment 24-well plates (Euroclone) and grown in suspension for the indicated days. Alternatively, embryoid bodies were transferred to adherent cell culture dishes and cultivated for the indicated time periods to obtain contractile cardiomyocyte structures. All used cells were tested for mycoplasma contamination in regular intervals.

**Viral transduction and generation of stable cell lines**. Retroviral vectors such as pLPC-24xMS2, pLPC-mOct4P4-24xMS2, pLPC-*mOct4P4*-deletion constructs pLPC-200 bp-*mOct4P4*-24xMS2, pLPC-(−200 bp)-*mOct4P4*-24xMS2, pPLC-*hOCT4P3*-24xMS2, pLPC-200bp-*hOCT4P3*-24xMS2, and pMSCV-HA-MS2-Flag were packaged using 293GP cells. Forty-eight hours post transfection 10 ml of supernatants were harvested, filtered through a 0.45 μm filter and used to infect OVCAR-3 or ES cells in presence of polybrene. Twenty-four hours later medium was replaced with selection medium to obtain stable cell pools. Lentiviral vectors pLX-sgOCT4P3, pLX-sg*Oct4P4*, and pHAGE-EF1α-dCas9-HA-KRAB (Addgene plasmid #50919) were packaged using 293T cells. Ten millilitre of supernatants were harvested, filtered through a 0.45 μm filter and used to infect OVCAR-3 or ES cells in presence of polybrene. Twenty-four hours later medium was replaced with selection medium to obtain stable cell pools. ES were transduced with retroviral vectors; OVCAR-3 cells with lentiviral and retroviral vectors. Cell lines infected with pLPC and pHAGE vectors were maintained in culture with 3 μg/mL of Puromycin and pMSCV and pLX vectors with 4 μg/mL of Blasticidine.

**Transient transfection of plasmids and siRNAs**. Transient transfections of plasmids were performed using TransIT®-LT1 transfection reagent (#MIR-2300, Mirus). Transient transfection of siRNA was performed using Lipofectamine RNAiMAX reagent (Invitrogen) according to manufacturer's suggestions. Following siRNAs were used for transient siRNA experiments: Fus: GCAA-CAAAGCUACGGACAA (Eurofins Genomics); Suv39h1: CCAAUUACCUGGUGCAGAA (Thermo Scientific Dharmacon); mOct4P4 GAGCAUGAGUGGAGAGGAA (Thermo Scientific Dharmacon). Control siRNA

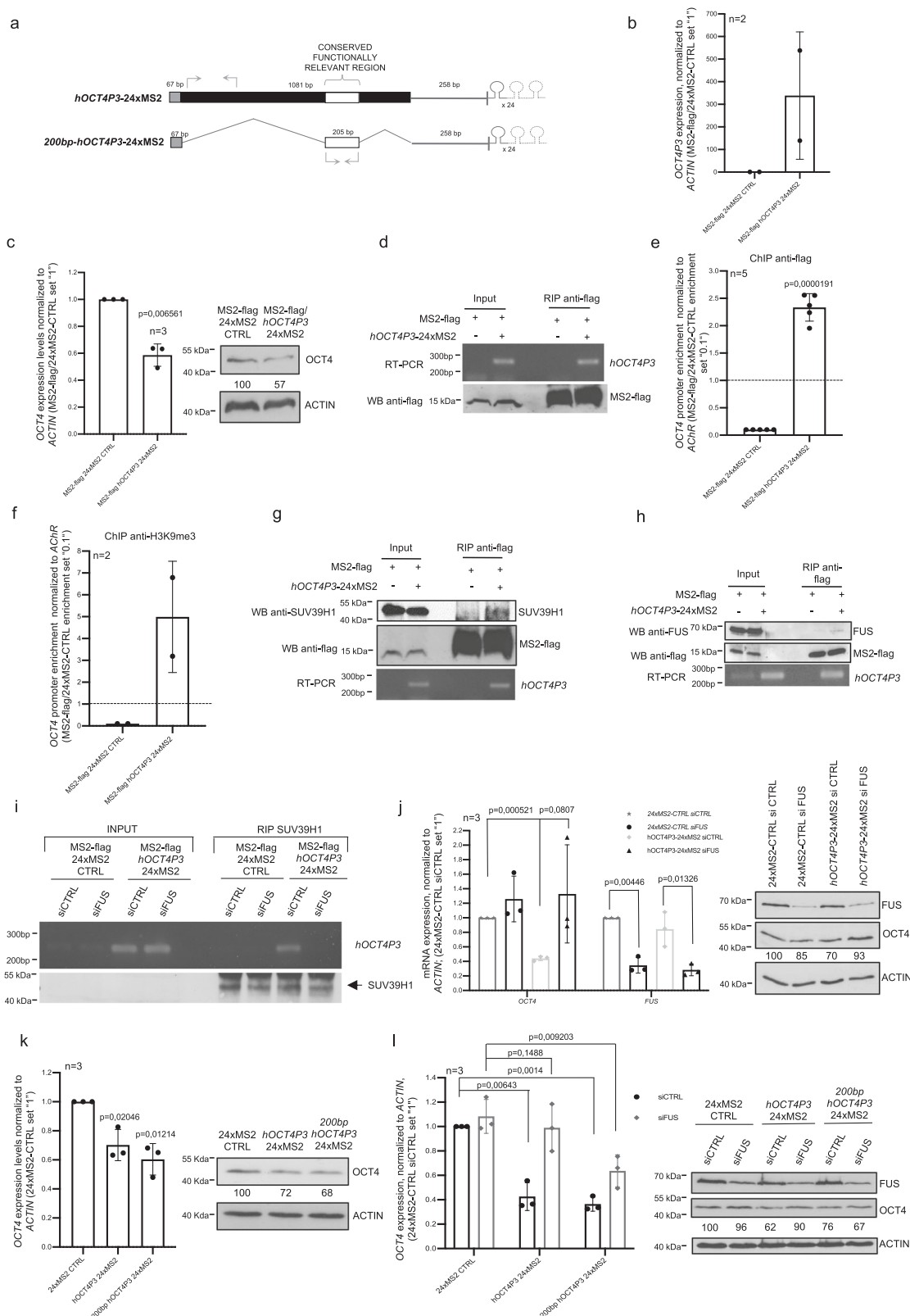

was used as a negative control: Non-Targeting siRNA#1, TAGCGACTAAACA-CATCAA (Thermo Scientific Dharmacon).

**RNA immunoprecipitation.** Experimental cells were scraped in RIPA buffer (50 mM Tris-Cl, pH 7.5, 1% Nonidet P-40 (NP-40), 0.5% sodium deoxycholate, 0.05% sodium dodecyl sulfate, 1 mM EDTA, 150 mM NaCl) and supplemented with protease inhibitors (Complete, Roche) and RNaseOUT (Invitrogen). After incubation at 4 °C for 20 min, cell lysates were centrifuged. The supernatant was precleared for 1 h at 4 °C with protein A/G PLUS-Agarose (protein A/G PLUS-Agarose—sc-2003; Santa Cruz Biotechnology supplemented with yeast tRNA. 0.1 mg/mL). The precleared supernatant was incubated overnight at 4 °C with rabbit polyclonal anti-TLS/Fus (ab23439, abcam), mouse monoclonal anti-KMT1A/Suv39h1 (2.5 mg/ml, ab12405, Abcam) or mouse monoclonal anti-FLAG M2, clone M2 (2.5 mg/ml; F1804; Sigma) antibodies. RNA–protein complexes were recovered with protein A/G PLUS-Agarose beads and were washed six times in RIPA buffer. An aliquot of beads containing immunoprecipitated samples were

**Fig. 7 Evolutional conservation of pseudogene lncRNA control of Oct4. a** Schematic representation of the OCT4P3-24xMS2 and 200 bp-OCT4P3-24xMS2 constructs. Length of pseudogene segments are indicated. Gray boxes, sequences with homology to *Oct4/OCT4* 5'UTR; gray lines, sequences with homology to *Oct4/OCT4* 3'UTR. White boxes, 200 nucleotide *hOCT4P3* region; 24xMS2 RNA stem loops aer indicated; arrows, position of RT-PCR primers. **b** Expression values of *hOCT4P3* in *hOCT4P3*-24xMS2 OVCAR-3 cells, as determined by qRT-PCR. ACTIN was used as reference. **c** OCT4/OCT4 expression in 24xMS2-CTRL and *hOCT4P3*-24xMS2 OVCAR-3 cells, as determined by qRT-PCR (left panel) and western blotting (right panel). ACTIN/ ACTIN was used to normalized expression values in qRT-PCR and as loading control in western blotting experiments, respectively. Numbers represent OCT4/ACTIN ratio (24xMS2-CTRL was set "100"). **d** Anti-flag RIP experiments in control and MS2-flag/*hOCT4P3*-24xMS2 OVCAR-3 cells. Top panel, agarose gel electrophoresis after *hOCT4P3* specific, quantitative RT-PCR; bottom panel, western blotting of RIP eluates using anti-flag specific antibodies. **e**, **f** Evaluation of enrichment of flag-MS2 (**e**) and H3K9me3 (**f**) at the *OCT4* promoter in control and *hOCT4P3*-24xMS2 OVCAR-3 cells by ChIP followed by quantitative RT-PCR. **g**, **h** Anti-flag RIP in control and *hOCT4P3*-24xMS2 OVCAR-3 cells. Top panels, western blotting of RIP eluates using the indicated antibodies. Bottom panel, agarose gel electrophoresis after *hOCT4P3* specific, qRT-PCR. **i** Anti-SUV39H1 RIP in control and *hOCT4P3*-24xMS2 OVCAR-3 cells after siRNA mediated depletion of FUS. Top panel, *hOCT4P3* specific qRT-PCR followed by agarose gel electrophoresis; bottom panel, anti-SUV39H1 western blotting using RIP eluates. **j** OCT4 and FUS expression in control and *hOCT4P3*-24xMS2 overexpressing OVCAR-3 cells after siRNA-mediated depletion of FUS, as determined by qRT-PCR (left panel) and western blotting (right panel). ACTIN was used as loading control. Numbers represent OCT4/ACTIN ratio as mean of three independent experiments (24xMS2-CTRL siCTRL was set "100"). **k** OCT4 expression in 24xMS2-CTRL, *hOCT4P3*-24xMS2 and 200 bp-*hOCT4P3*-24xMS2 OVCAR-3 cells as determined by qRT-PCR (left panel) and western blotting (right panel). ACTIN was used to normalized OCT4 expression values in qRT-PCR and western blotting experiments. Numbers represent OCT4/ACTIN ratio (24xMS2-CTRL was set "100") (right panel). **l** OCT4 expression levels in 24xMS2-CTRL, OCT4P3-24xMS2, and 200 bp-*hOCT4P3*-24xMS2 OVCAR-3 cells under control or FUS knockdown condition, as determined by qRT-PCR (left panel) and western blot (right panel). ACTIN was used as loading control. Numbers represent OCT4/ACTIN ratio as mean of three independent experiments (24xMS2-CTRL siCTRL was set "100"). Error bars represent standard deviation. Precise *p* values are indicated. *n* number of independent experiments carried out.

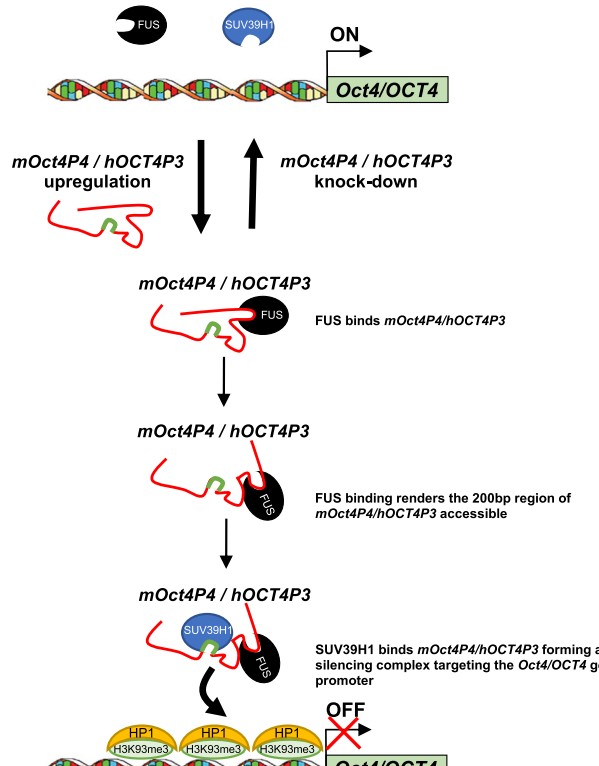

**Fig. 8 A model of FUS-dependent licensing of targeted silencing of *OCT4* by *OCT4* pseudogene ncRNAs.** FUS binds to endogenous, full length *mOct4P4/hOCT4P3* to provide access for SUV39H1 to 200 nucleotide lncRNA region (highlighted in green). Binding of *mOct4P4/hOCT4P3* lncRNA to SUV39H1 creates a silencing complex with target specificity for the promoter of the ancestral *Oct4/OCT4* leading to repression of parental *Oct4/OCT4* by creating local H3K9me3 containing heterochromatin.

saved for western blotting analysis. Remaining beads were used to obtain immu-noprecipitated RNA that was analyzed by qRT-PCR. A mouse monoclonal anti-HA antibody, clone HA-7 (2.5 mg/ml, Sigma H9658) was used as negative control for immunoprecipitation.

**Statistics and reproducibility**. A one-tailed *t* test was performed to calculate *p* values and statistical significance was set at *p* < 0.05. Each finding was confirmed by three independent biological replicates, unless differently specified. Error bars represent standard deviation.

**Reporting summary**. Further information on research design is available in the Nature Research Reporting Summary linked to this article.

## Data availability
All data generated or analyzed during this study are included in this published article and related Supplementary information files. Source data of blots and gels are shown in Supplementary Fig. 6.

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

## Acknowledgements

M.S. and E.C. are supported by AIRC post-doctoral fellowships. M.R. is enrolled in the PhD program for Molecular Medicine at the University of Trieste. *Financial support:* This work was supported by AIRC grants (Rif 17756 to R.B., Rif 18381 to S.S., Rif. 22174, and Rif. 22759 to G.D.S.) and PRIN bando 2017 to G.D.S.

## Author contributions

M.S., S.S., and R.B. designed the experiments; M.S., E.C., and M.R. carried out the experiments; G.D.S. and C.S. supported the experimental design and data interpretation; S.S. and R.B. wrote the paper.

## Competing interests

The authors declare no competing interests.
