## [Peer Review File · Communications Biology]

Reviewers' comments:

Reviewer #1 (Remarks to the Author):

In the manuscript submitted by Scarola et al., they propose that a transcript produced by a noncoding pseudogene for OCT4 is expressed, recruited to the promoter for the OCT4 gene and negatively impacts OCT4 expression in a manner that depends on the transcription regulators FUS and SUV39H1. The work is well written, and a substantial amount of experimental data is provided and presented in a fairly straightforward manner.

Nevertheless, the data falls considerably short of demonstrating the proposed mechanism and supporting this mode of regulation to be specific for the OCT4 gene or the proteins suggested. A fair amount of the data does not really address the model in any direct way. Those critical experiments that are provided and required, have not received the attention necessary to draw the conclusions made. Comments below focus on the missing data or those data that require more attention to be able to support the claims made.

Specific Comments:

1. It's not entirely clear how the experiment for Fig 1B is done. However, it is essential that the copy number of transcripts per cell be determined to claim a trans mechanism and compared to a control of a mRNA or ncRNA with a well-known range for numbers of transcripts per cells. Are the number of transcripts per nucleus 90% of only 2 molecules per cell or 200 molecules per cell? The proteins suggested to be involved, FUS and SUV39H1, have little to no inherent specificity but found at least at half or more of gene promoters (e.g. work from the Cech, Reed, Alberti, and Ohno labs). Such as it is, the lncRNA gene will be required to provide a considerable number of molecules to find and maintain any significant occupancy at the OCT4 promoter. One would also expect such levels to be visible by northern as for an mRNA, whose expression tends to be in the range of dozens of molecules per cell.
2. Since the Fig 1F suggests the lncRNA to be a regulator but not the primary or dominant regulator, relative comparisons of transcript levels by realtime PCR does not offer the best support for the claim of mechanism or biological significance. The mechanism is presumed to involve transcription, which can be and often is considerably uncoupled from steady-state mRNA levels through a number of post-transcriptional regulatory points. The biological significance must start at protein levels, which means having westerns to accompany the realtime PCR is important.
3. The crux of this story rests on the reliability of ChIP-PCR data. This would warrant that more than one or two primers be used to sample the location of signals along the chromatin. The authors appear to use a control DKK primer set for normalization. However, showing enrichment for the promoter over adjacent regions along the OCT4 gene can reasonably be expected. And as stated above, the authors should be able to choose from a sizeable list of genes to find FUS and SUV39H1 and test whether a claim for specificity is valid for OCT4.
4. Since the promoter is implicated, it is reasonable to expect data to be shown for RNA Pol II to accompany these ChIP experiments. FUS has been shown by several labs to be one of the most abundant proteins bound to RNA Pol II, so is FUS moving on and off the gene because the polymerase moves or because of the lncRNA? Since RNA Pol II levels for a promoter alone can be misleading and transcription pausing can possibly increase promoter levels, a functional assay such as run-on assays could possibly substitute.
5. Can the authors explain why the extreme overexpression of the lncRNA driven by a CMV promoter cannot have a better effect on OCT4 transcription than that of the endogenous transcript in Fig 1? Is OCT4 expression in those cells already considerably lower than the cell lines feature in Fig 1?
6. The majority of cell data for FUS argues it is bound to most RNA transcripts, particularly mRNAs. Any claims to date for FUS specificity to an RNA sequence/structure beyond what has been called "degenerate" specificity have not stood up to rigorous scrutiny. As such, it's hard to see what the significance of the RIP data is. The signals are quite weak and if the claim is that

recruitment to the OCT4 promoter is "licensed" by FUS and SUV39H1, it seems parts C thru F on Fig 5 should include the -200 construct, in order to claim "specificity", at least on the functional level.

7. Another critical problem is that the co-IP data for FUS and SUV39H1 doesn't appear to have worked. The authors should at least show these whole gels in Supp Fig 2A and B but what is shown do not look like bands. This critical experiment needs to be in the main body. There is now a substantial amount of proteomics data for the FUS interactome, has SUV39H1 ever been noted? Is this unique to these cells?

8. The authors need to include the rest of their MS data for the lncRNA IP and its controls. The Supp Table 1 is not easy to understand and it's unclear why FUS would be listed as a MW of 14 kDa? How enriched is FUS over the controls? And SUV39H1? Last, what else can be candidates for being enriched for this overexpressed lncRNA?

9. Since the last decade has seen an explosion in claims for lncRNAs to bind promoters in trans and few details for how a lncRNA can interact with a specific promoter have come forward, it would be most helpful if the authors were able to compare their levels of enrichment to any other published claim and note if their claim is superior to the bulk of the literature. Perhaps they can at least compare the levels of their lncRNA bound to the promoter to those of the nascent transcripts for OCT4. Presumably binding of the lncRNA should increase post-differentiation to ultimately exceed the amounts of nascent OCT4 transcript still being transcribed, as the gene shuts down. Without a better approach to establish the transcript's presence at a promoter, it's hard to see that the addition of nonspecific RNA-binding proteins to the mechanism has elevated this study much beyond the lab's 2015 publication. Perhaps the point is buried under those data that does not address the questions of specificity nor that transcription itself is affected, as opposed to regulation through some other cellular pathway.

Additional Comments:

1. The swapping of bars for nuclear and cytoplasmic RNA in figs 1B and 3C can be confusing.
2. Does Fig 1 then argue that the lncRNA must maintain high expression for OCT4 to be silenced and therefore found to be highly expressed in most cells not expressing OCT4?
3. Presumably the comparisons of enrichment by ChIP have negative values because they are log values. The authors should label the plots this way.
4. The excessive numbers of significance bars displayed in Fig 6 obscures what the authors feel the reader should focus on that best supports the claims about the specificity of the 200 bp region.
5. It's hard to determine nor is it very obvious from the legend what the significance values in Figs 2 and 3 are in reference to. A clearer emphasis of this point in the legend would help.

Reviewer #2 (Remarks to the Author):

Overall this new manuscript by Scarola et al. contains a lot of data, with well controlled experiments with appropriate statistical tests and fair conclusions accurately drawn from the data presented. Among other findings, the authors identify an OCT4 pseudogene mechanism of action and demonstrate this pseudogene is required for proper loading of the methyltransferase Suv39 to the Oct4 gene. The authors have run all the proper controls and the actual data presented in the figures are strong. Experimentally the paper is quite solid and well laid out in a logical manner. However, some more experimental details in the manuscript would be required to merit publication. Major and minor points in this regard below:

Major points:

1. Some of the experiments presented are fairly complex, for instance the formation of embryoid bodies. The M&M section should include more details on these experiments. Additionally, it would be nice to see images of the embryoid bodies during this experiment (either in the primary figures or supplement), so the reader can see what the cells look like @T0, and D3-10.

2. More experimental details on how actual experiments were conducted should be in the materials section. For instance, there is no description of how ChIP was conducted, how virus was packaged (packaging vectors used), how virus was concentrated or used neat, how much reagents were used for experiments such as ug DNA used in transfection and volume or MOI of virus used in infections.

3. Paragraph structure could be improved. For instance:

a. The introduction is one long paragraph. This could be broken down into separate paragraphs to separate out focus areas, more typically introductions are 3-5 paragraphs setting the tone for the manuscript and giving appropriate background to appreciate the work. For example, more details about the known roles of FUS would be helpful in understanding the importance of this manuscript in context to the field.

b. The results sections are all one paragraph with sometimes a separate thought as a single sentence paragraph. These also could be expanded to better go through the large amount of experimental data in this manuscript.

Minor points:

4. The title contains a lot of acronyms that are not defined, for instance HMTase could be removed and the title would have the same meaning.

5. Gene names are sometimes capitalized and sometimes first letter in caps. This should be consistent throughout the document. For instance, the Oct4 gene is written as "OCT4" and "Oct4" in the abstract. It should be first letter caps, and preferable in italics when referring to the gene name throughout.

6. Paper would benefit from additional proofreading for proper use of gene names, grammar, and paragraph structure.

Reviewers' comments:

Reviewer #1 (Remarks to the Author):

In the manuscript submitted by Scarola et al., they propose that a transcript produced by a noncoding pseudogene for OCT4 is expressed, recruited to the promoter for the OCT4 gene and negatively impacts OCT4 expression in a manner that depends on the transcription regulators FUS and SUV39H1. The work is well written, and a substantial amount of experimental data is provided and presented in a fairly straightforward manner.

Nevertheless, the data falls considerably short of demonstrating the proposed mechanism and supporting this mode of regulation to be specific for the OCT4 gene or the proteins suggested. A fair amount of the data does not really address the model in any direct way. Those critical experiments that are provided and required, have not received the attention necessary to draw the conclusions made. Comments below focus on the missing data or those data that require more attention to be able to support the claims made.

We thank the reviewer for the positive comments on our manuscript in particular stating that our story “*is well written, and a substantial amount of experimental data is provided and presented in a fairly straightforward manner*”. We really appreciate all the suggestions and comments that were very helpful in further improving the quality and the claims of our manuscript.

Specific Comments:

1. It's not entirely clear how the experiment for Fig 1B is done. However, it is essential that the copy number of transcripts per cell be determined to claim a trans mechanism and compared to a control of a mRNA or ncRNA with a well-known range for numbers of transcripts per cells. Are the number of transcripts per nucleus 90% of only 2 molecules per cell or 200 molecules per cell? The proteins suggested to be involved, FUS and SUV39H1, have little to no inherent specificity but found at least at half or more of gene promoters (e.g. work from the Cech, Reed, Alberti, and Ohno labs). Such as it is, the lncRNA gene will be required to provide a considerable number of molecules to find and maintain any significant occupancy at the OCT4 promoter. One would also expect such levels to be visible by northern as for an mRNA, whose expression tends to be in the range of dozens of molecules per cell.

REPLY TO REVIEWER:

We thank the reviewer to address the interesting issue of a possible link between lncRNAs function and lncRNAs expression levels.

Along these lines, recent studies on mammalian X-chromosome inactivation driven by the lncRNA *Xist* revealed that approximately 100-200 lncRNA molecules are sufficient to cover the entire X chromosome. This corresponds to only 1 molecule of RNA per 1Mb of DNA, highlighting the power of lncRNAs in chromatin regulation (**PMID: 26195790**; **PMID: 25057298**, **PMID: 31795917**). As suggested by the reviewer, we provide new data determining RNA abundance of *hOCT4P3*, *OCT4* and a reference gene *DAXX* (a replication independent histone chaperon).

To give information on *hOCT4P3* transcript levels we have performed a quantitative RT-PCR based approach, previously used to determine *mOCT4P4* lncRNA copy numbers in mouse mESCs and pMEFs (Scarola et al. Nature Communications 2015).

Briefly, a limited dilution series of vectors containing subcloned fragments of *hOCT4P3*, *DAXX* or *OCT4* were used as internal standard for the quantitative determination of *OCT4*, *DAXX* or *OCT4P3* RNA molecules (for detailed description of procedure, see Supplementary Information section of the revised version of the manuscript). *hOCT4P3* RNA numbers in Ovar3 were found to be approximately $7,86 \times 10^4$ molecules/ μg reverse transcribed RNA. *OCT4* and *DAXX* mRNA were found at a concentration of $6,72 \times 10^4$ and $10,2 \times 10^6$ molecules/ μg reverse transcribed RNA, respectively. We thus estimate that *DAXX* mRNA levels are ca. 130 fold higher than *hOCT4P3* and 150 fold higher than *OCT4* in Ovar-3.

Typically, mammalian cells contain approximately 30 pg total RNA/cells. Consequently, our calculations suggest that OVCAR-3 cells may contain approximately 2 molecules of *OCT4* and 2,5 molecules of *hOCT4P3* per cell.

In conclusion, the *hOCT4P3* lncRNA fall into the class of rare or low-abundance mRNAs/lncRNAs that are reported to have a copy number of only 2–15 molecules per cell. As mentioned above, recent reports showed that 1 molecule of *Xist* lncRNA per 1Mb of DNA is sufficient to mediate chromosome wide silencing. This supports that *hOCT4P3*, although only present at low copy number, has an important role in parental gene expression control (**PMID: 26195790**; **PMID: 25057298**, **PMID: 31795917**).

We present this data in the **new Supplementary Fig. 1A** and the results section at **page 8, lines 18 to 21**: “Quantitative RT-PCR experiments revealed that *hOCT4P3* and *OCT4* transcript levels are 130 or 150 fold lower than the housekeeping gene *DAXX*. However, although present at low copy number, *hOCT4P3* has an important role in parental gene expression control.

Low *hOCT4P3* RNA and *OCT4* mRNA transcript numbers in OVCAR-3 cells render it very difficult to visualize the transcripts in Northern blotting experiments. Following the reviewer’s suggestions, we made various attempts to detect the endogenous RNAs with oligo-probes and also longer, body labelled probes. However, these experiments failed to detect *hOCT4P3* lncRNA. We attribute this to the low copy number of transcripts, but also to the fact that also other (only poorly characterized) *hOCT4* pseudogene lncRNAs (with different size) are also expressed in OVCAR-3 cells. These transcripts show high level of sequence identify with *hOCT4P3* and thus compete for the used probes. This problem can be avoided by the use of highly specific oligos in RT-PCR experiments as used throughout the study.

2. Since the Fig 1F suggests the lncRNA to be a regulator but not the primary or dominant regulator, relative comparisons of transcript levels by realtime PCR does not offer the best support for the claim of mechanism or biological significance. The mechanism is presumed to involve transcription, which can be and often is considerably uncoupled from steady-state mRNA levels through a number of post-transcriptional regulatory points. The biological significance must start at protein levels, which means having westerns to accompany the realtime PCR is important.

REPLY TO REVIEWER:

Demonstrating pseudogene lncRNA function also on the protein level of the target gene is highly relevant – we thank to reviewer to stress this issue.

In the revised version of the manuscript we have added a western blot for OCT4 using extracts derived from embryoid bodies with altered *mOct4P4* lncRNA expression. We show that at day 10 of embryoid body formation, OCT4 protein levels are maintained at high levels in dCas9 sgOct4P4 cells when compared to dCas9 control EBs. This data can be found as **new Supplementary Fig. 1B** in the revised version of the manuscript. We mentioned this data in the revised text at **line 32 pag 8**.

3. The crux of this story rests on the reliability of ChIP-PCR data. This would warrant that more than one or two primers be used to sample the location of signals along the chromatin. The authors appear to use a control DKK primer set for normalization. However, showing enrichment for the promoter over adjacent regions along the OCT4 gene can reasonably be expected. And as stated above, the authors should be able to choose from a sizeable list of genes to find FUS and SUV39H1 and test whether a claim for specificity is valid for OCT4.

REPLY TO REVIEWER:

We thank the reviewer for this critical comment. In our previous study on murine *mOct4P4* lncRNA we have used 2 primer pairs that amplify different *Oct4* promoter regions in ChIP experiments (Scarola et al. 2015). As suggested by the reviewer, we have now included additional primer pairs that amplify genomic adjacent regions located up and downstream of the mouse *Oct4* promoter. Importantly, using the *Dkk* gene as reference we were able to confirm enrichment of the *mOct4P4-MS2* lncRNA at the *Oct4* promoter of flag-MS2 overexpressing mESCs, but were not able to detect a specific recruitment to the promoter of the *Daxx*, *H2q10*, *Ceher1*, *Pp1r18* and *Rab5A* gene promoters. This data confirms thoroughly that *mOct4P4* is specifically recruited to the Oct4 promoter.

This new data can be found as **new Supplementary figure 2C**. We refer to these new data in the results section at **page 10, lines 13-16**: “Of notice, ectopic *mOct4P4-24xMS2* was exclusively recruited to *Oct4* promoter the but not to the promoters of *Daxx*, *H2q10*, *Ceher1*, *Pp1r18* and *Rab5a* that are localized up- and downstream of *Oct4* (Supplementary figure 2C).”

4. Since the promoter is implicated, it is reasonable to expect data to be shown for RNA Pol II to accompany these ChIP experiments. FUS has been shown by several labs to be one of the most abundant proteins bound to RNA Pol II, so is FUS moving on and off the gene because the polymerase moves or because of the lncRNA? Since RNA Pol II levels for a promoter alone can be misleading and transcription pausing can possibly increase promoter levels, a functional assay such as run-on assays could possibly substitute.

REPLY TO REVIEWER:

We thank the reviewer for this very valid comment and proposal to investigate in depth the role of FUS in mediating reduced *Oct4* mRNA levels. In fact, initially we were hypothesizing that FUS might have a central role in silencing the promoter. However, our experimental data (shown in our manuscript) indicated to us that we needed to change our first model for the following reasons:

- Figure 4F shows that siRNA mediated depletion of *mOct4P4* is linked with a loss of binding of FUS to the promoter of the parental *Oct4* gene. Thus, *mOct4P4* lncRNA - rather than RNA Pol II – appears crucial to bring FUS to the *Oct4* promoter. We highlight this point in the discussion section on **page 16, lines 20-21** stating that: “*Thus, we conclude that FUS does not have a central role in closing the OCT4 promoter*”.
- In addition, Fig.6F and Fig. 7L show that FUS is dispensable for silencing of Oct4 by the 200-nucleotide minimal sufficient *mOct4P4/hOCT4P3* versions. This data favors a model where FUS has a role in “preparing” the full length *mOct4P4/hOCT4P3* lncRNA to be ready for recruiting SUV39H1 by the 200-nucleotide minimal region.

However, we completely agree that a detailed characterization of molecular events occurring at the *Oct4* promoter are of high relevance for future studies. Such work can include RNA Pol II CTD phosphorylation status, quantitative *mOct4P4/hOCT4P3* lncRNA studies, run-on assays, CAGE tagging studies in cell model systems established by us.

We refer to the issue in the discussion section of the revised version of the manuscript on **page 16, lines 30-32**: “*In addition, the impact of Oct4 promoter associated pseudogene transcripts on transcriptional initiation and promoter evasion remain to be addressed.*”

5. Can the authors explain why the extreme overexpression of the lncRNA driven by a CMV promoter cannot have a better effect on OCT4 transcription than that of the endogenous transcript in Fig 1? Is OCT4 expression in those cells already considerably lower than the cell lines feature in Fig 1?

REPLY TO REVIEWER:

hOCT4P3 loss of function experiments in OVCAR3 CRISP/dCas9 cells (Fig. 1I-K) demonstrate the importance of *hOCT4P3* lncRNA in silencing OCT4: a 60% reduction of *hOCT4P3* lncRNA is sufficient to mediate a reproducible, 2- and 1.5 fold increase in OCT4 mRNA and protein expression, respectively. This indicates that *hOCT4P3* mediated suppression of OCT4 is operative in experimental control OVCAR-3 cells. As pointed out by the reviewer, ectopic expression of *hOCT4P3* lncRNA further decreases OCT4 expression, is however not able to completely silence the OCT4 promoter (Fig. 1C, D). We speculate that saturation of pseudogene lncRNA mediated silencing of OCT4 may be reached already by moderate OCT4 pseudogene overexpression. However, it is also essential to consider that expression of the pseudogene lncRNA from an ectopic site may result in OCT4 silencing with reduced efficiency, when compared to a pseudogene lncRNA expressed from the endogenous locus.

These critical issues can be solved by inserting an inducible promoter upstream of the *hOCT4P3* pseudogene; these experiments are laborious and time intensive and go beyond the time-scale for this manuscript.

However, we want to underline that data from *mOct4P4* and *hOCT4P3* CRISP/dCas9 cells as well as lncRNA knock-down experiments (Scarola et al. 2015 and this manuscript) clearly demonstrate that OCT4 expression is regulated by its offspring pseudogene lncRNAs.

6. The majority of cell data for FUS argues it is bound to most RNA transcripts, particularly mRNAs. Any claims to date for FUS specificity to an RNA sequence/structure beyond what has been called “degenerate” specificity have not stood up to rigorous scrutiny. As such, it’s hard to see what the significance of the RIP data is. The signals are quite weak and if the claim is that recruitment to the OCT4 promoter is “licensed” by FUS and SUV39H1, it seems parts C thru F on Fig 5 should include the -200 construct, in order to claim “specificity”, at least on the functional level.

REPLY TO REVIEWER:

Figure 5A and B in the original version of the manuscript demonstrate that the -200 construct (lacking the functionally silencing-relevant sequence stretch) DOES interact with FUS, but DOES NOT bind SUV39H1. We therefore feel that including data on the -200 construct in panel C and D is redundant. However, we completely agree with the reviewer that is essential to better demonstrate that FUS has relevant specificity/affinity for the *mOct4P4* transcript. We have therefore extended our anti-FUS and anti-SUV39H1 RIP experiments to other control RNAs. In the revised version of our manuscript we show in **modified Figure 5E and F** that *FUS and SUV39H1 bind mOct4P4 lncRNA but DO NOT bind Oct4, Sox2, or Nanog mRNAs and also DO NOT bind the highly abundant Gapdh and Actin transcripts*. This demonstrates that under our experimental conditions FUS and SUV39H1 have a clear affinity towards *mOct4P4* lncRNA, but do not bind unrelated RNAs. We refer to this result in the results section of the manuscript on **page 13, lines 13-16** “*Importantly, RIP experiments using mESCs demonstrated that under our experimental conditions SUV39H1 and FUS exclusively bind mOct4P4 lncRNA but not Oct4 or other mRNAs such as Sox2, Nanog, Gapdh or Actin*”

As suggested by the editor, we were toning down the statement of FUS dependent binding of SUV39H1 to OCT4 pseudogene lncRNA. We state now on the results section on **page 15, Lines 22-24** that: “*FUS binding*

to the endogenous mOct4P4/hOCT4P3 lncRNA plays an important role in rendering the 200-nucleotide region accessible for SUV39H1 binding.

(original text: “FUS binding to the endogenous mOct4P4/hOCT4P3 lncRNA is required to render the 200-nucleotide region accessible for SUV39H1 binding.”)

7. Another critical problem is that the co-IP data for FUS and SUV39H1 doesn't appear to have worked. The authors should at least show these whole gels in Supp Fig 2A and B but what is shown do not look like bands. This critical experiment needs to be in the main body. There is now a substantial amount of proteomics data for the FUS interactome, has SUV39H1 ever been noted? Is this unique to these cells?

REPLY TO REVIEWER:

Following the suggestion of the reviewer, we have repeated the IP experiment, also optimizing gel running conditions to allow better separation of antibody heavy chain from the SUV39H1 protein band. Confirming original data, we did not find evidence for FUS - SUV39H1 interaction in IP experiments. As this experiment does only show “non-interaction” we decided to leave the figure in the supplementary data section. However, as requested by the reviewer, we show now the entire western blotting membranes both for original and new experiments. These data can be found in modified **Supplementary figure 3A, B**.

Importantly, lack of SUV39H1 in published data on the FUS interactome is in line with our results (PMID: 27460707; PMID: 29884807; PMID: 31693373; PMID: 27164932; Reber et al. 2019, bioRxiv 806158). This comment can be found on **page 12, line 27** of the modified version of our manuscript: “*This is in line with a lack of SUV39H1 in published data on the FUS interacting proteome*”

8. The authors need to include the rest of their MS data for the lncRNA IP and its controls. The Supp Table 1 is not easy to understand and it's unclear why FUS would be listed as a MW of 14 kDa? How enriched is FUS over the controls? And SUV39H1? Last, what else can be candidates for being enriched for this overexpressed lncRNA?

REPLY TO REVIEWER:

Comment to coverage of peptides (14 kDa FUS):

The peptide library of the data analysis software contains the full length peptide sequence of FUS but also a panel of FUS fragments. The software assembles peptide reads to cover the individual peptides present in the library, prioritizing high % matches. Thus, shorter peptides (such as the 14 kDa FUS peptide in the library) can reach higher coverage and will be prioritized. We agree that the presentation of the original table is misleading. We therefore eliminated the column “No peptides matched” from the original table. In order to provide full information we have added the complete mass spectrometry data as **new Supplementary data excel file**.

Comment to controls and FUS enrichment:

- As we have exclusively analyzed protein bands that only appear in *mOct4P4* lncRNA RIP eluates, we cannot provide data for a control sample. However, we were not able to identify FUS derived peptides in the other gel slices analyzed by mass spectrometry, providing additional information on the specificity of this highly abundant protein in binding *mOct4P4* lncRNA (see **new Supplementary data file**) I want to underline that separate anti-FUS RIP experiments validated the interaction of mOct4P4 with FUS (Fig. 4A). Thus, FUS is a “real” interactor of mOct4P4 lncRNA (Fig. 2A-C).
- When aiming to identify *mOct4P4* lncRNA interacting proteins by RNA pulldown experiments, we cut out protein bands that - judged by visual inspection - exclusively appeared in lanes of *mOct4P4-24xMS2* lncRNA pull down eluates (Fig. 4A). Doing so, we did not include gel-slices in the kDa range of SUV39H1 (47 kDa); thus we cannot provide mass spec data from these control samples. However, we have re-visited all peptides identified in analyzed samples. We did not find evidence for presence of SUV39H1 peptides.
- We also identified proteins like NCL, eIF4B, HMGB1/2 and PPIB as potentially functionally relevant *mOct4P4* lncRNA interactors; however, their role remains to be demonstrated. We have inserted on **Page 16, lines 14-15** of the discussion section the following statement: “*RNA pull-down experiments revealed a series of mOct4P4 lncRNA interacting proteins with a potential role in silencing parental Oct4*”.

As requested by the reviewer we have added the entire mass spectrometry data set as **new Supplementary data excel file** and refer to this data on **page 12, line 25** of the results section.

9. Since the last decade has seen an explosion in claims for lncRNAs to bind promoters in trans and few details for how a

lncRNA can interact with a specific promoter have come forward, it would be most helpful if the authors were able to compare their levels of enrichment to any other published claim and note if their claim is superior to the bulk of the literature. Perhaps they can at least compare the levels of their lncRNA bound to the promoter to those of the nascent transcripts for OCT4. Presumably binding of the lncRNA should increase post-differentiation to ultimately exceed the amounts of nascent OCT4 transcript still being transcribed, as the gene shuts down. Without a better approach to establish the transcript's presence at a promoter, it's hard to see that the addition of nonspecific RNA-binding proteins to the mechanism has elevated this study much beyond the lab's 2015 publication. Perhaps the point is buried under those data that does not address the questions of specificity nor that transcription itself is affected, as opposed to regulation through some other cellular pathway.

REPLY TO REVIEWER:

Comment to transcript presence at OCT4 promoter:

We agree that the determination of molecular promoter events and nascent *Oct4-mOct4P4* transcript kinetics can give important insights into the spatio-temporal action of the pseudogene lncRNA at the *Oct4* promoter. However, we feel that these experiments go beyond the goal of this manuscript that focusses on the functional analysis of central factors in pseudogene lncRNA mediated silencing of parental gene expression. Addressing this specific question would open many questions and require a large set of specialized methods. We completely agree with the reviewer on the relevance of these experiments and state in the discussion section of the revised version of the manuscript on **page 16, lines 30-32**: "*In addition, the impact of Oct4 promoter associated pseudogene transcripts on transcriptional initiation and promoter evasion remains an interesting open issue to be addressed.*"

In order to address the reviewer's suggestion to provide more evidence for the specificity of *mOct4P4* lncRNA for the *OCT4* promoter, we applied an alternative experimental approach. In particular, anti-flag-MS2 ChIP on control and *mOct4P4-24xMS2* lncRNA expressing cells followed by quantitative RT-PCR demonstrated *mOct4P4-24xMS2* lncRNA enrichment at the *OCT4* promoter, but absence of the *mOct4P4* lncRNA on promoters of 5 neighboring genes (*Daxx*, *H2q10*, *Cchcr1*, *Pp1r18* or *Rab5a*) along Chr.17. This data is shown in **new Supplementary Fig. 2C**

In point 9 the reviewer states that the current study, lacking an approach to establish the transcript's presence at a promoter, may not go sufficiently beyond the data published in our past Nature Communications paper (2015) that discovered a role for the *mOct4P4* lncRNA in recruiting to the parental *Oct4* gene promoter *in trans*. The current study has the goal to shed light into the molecular information the *Oct4* pseudogene lncRNA holds to mediate *Oct4* gene silencing. To address this issue, we have used a wide set of methods that comprise the generation of CRISPR/dCas9 model systems that allow the silencing of *Oct4P4* and *OCT4P3* in mouse and human cells, a complex RNA tagging system in combination with a deletion analysis and protein discovery by pull down experiments using MS2 tagged *Oct4* lncRNA.

We feel that this is a competitive experimental set-up that allowed to unravel important and new aspects pseudogene lncRNA mediated silencing of *Oct4* gene, as summarized below:

- CRISPR/dCas9 mediated silencing of *mOct4P4/hOCT4P3* expression significantly increased *Oct4/OCT4* expression. In mESCs, this led to the maintenance *Oct4* expression during embryonic stem cell differentiation, paired with the aberrant maintenance of self-renewal makers. This experiment validates – without affecting DNA sequence of the *mOct4P4/hOCT4P3* pseudogene locus – *mOct4P4/hOCT4P3* lncRNA function and its biological relevance in a key model for stem cell differentiation. These experiments were essential to validate loss of function data in the Nature Communications paper, that were based on siRNA mediated depletion of *mOct4P4*.
- A deletion analysis identifies a 200-nucleotide conserved region in the *mOct4P4* and *hOCT4P3* lncRNA required for *Oct4* promoter targeting and Suv39H1 mediated silencing. This demonstrates the i) biological function is limited to a short sequence element and ii) an excellent example for the formation of a new functional RNA element evolving from the paternal, protein coding *OCT4* transcript.
- A new *Oct4P4/OCT4P3* lncRNA interactor – FUS – has a critical role for the function of full length *mOct4P4/hOCT4P3*, unravelling a remarkable 2 step process lncRNA function during *Oct4* pseudogene silencing.
- A precise characterization of *Oct4* pseudogene lncRNA – protein interactions and promoter targeting.
- Specific binding of *mOct4P4* lncRNA to the *Oct4* promoter (and no binding to off-target gene promoters)
- Our data further demonstrate functional conservation of "our pseudogene lncRNA" in mouse and human cells – a key indicator for evolutionary relevance of pseudogene derived lncRNAs.

Additional Comments:

1. The swapping of bars for nuclear and cytoplasmic RNA in figs 1B and 3C can be confusing.

REPLY TO REVIEWER:

In the revised version of the manuscript we have changed color code of Fig 1B accordingly to Fig. 3C.

2. Does Fig 1 then argue that the lncRNA must maintain high expression for OCT4 to be silenced and therefore found to be highly expressed in most cells not expressing OCT4?

REPLY TO REVIEWER:

This is a good question – we would like to refer first to results from our previous work on *mOct4P4* (Scarola et al. 2015).

- In mESCs defined by high OCT4 and low *mOct4P4* expression, knock down of *mOct4P4* leads to further increased OCT4 expression (Scarola et al. 2015).
- Impairing the programmed increase of *mOct4P4* expression during mESC differentiation results in the maintenance of OCT4 expression in differentiated mESCs (Scarola et al. 2015).
- In pMEFs characterized by a lack of OCT4 expression and high *mOCT4P4* expression, knock-down of *mOct4P4* leads to a remarkable re-expression of OCT4 (Scarola et al. 2015).

Our new CRISP/dCas9 data show that already a 60% reduction of endogenous *hOCT4P3/mOct4P4* expression leads to increased *OCT4/Oct4* expression in OVCAR-3 cells and mESCs (Fig.1D, I-K); and impaired OCT4 silencing during EB differentiation (Fig. 1D, E-H).

Thus, in all used model systems, irrespective of *OCT4* or *OCT4*-pseudogene expression levels, endogenous *hOCT4P3/mOct4P4* has a suppressive effect on *OCT4* expression. However, our data indicate that enforced pseudogene expression – as a consequence of a differentiation or signaling event - enhances the suppressive effect on *OCT4* expression. This suggest that increasing *OCT4*-pseudogene expression results to reduced *OCT4* expression.

Work to address a possible linear relation between *OCT4* and *hOCT4P3* expression is currently investigated in the laboratory using fresh ovarian cancer patient specimen. However, to-date the number of enrolled patients is too low to allow reliable conclusions.

To address the reviewer's comment, we have inserted into **page 9, lines 14-16** the following phrase: *“Importantly, data from dCAS9-HAKRAB loss of function models also demonstrate that endogenous *mOCT4P4* and *hOCT4P3* lncRNAs have a suppressive action on the *Oct4/OCT4* promoter in mESCs and OVCAR-3 cells.”*

3. Presumably the comparisons of enrichment by ChIP have negative values because they are log values. The authors should label the plots this way.

REPLY TO REVIEWER:

A negative value indicates “x-fold reduction”, compared to the respective control. This is also indicated by the Y-axis labeling. I feel that this representation explains a reduction in chromatin compaction in an intuitive manner. We have revisited the labelling of all y-axes to prevent miss-interpretation by readers.

4. The excessive numbers of significance bars displayed in Fig 6 obscures what the authors feel the reader should focus on that best supports the claims about the specificity of the 200 bp region.

REPLY TO REVIEWER:

The p-values shown in Figure 6 indicate the significance of changes of experimental cells to all possible control conditions. As suggested, we tried to limit numbers of significance but finally we feel that is important to offer as much information as possible on the significance of all observed changes to the readers of the manuscript. This also supports the strength of developed models and experimental set-up. Therefore, we finally opted to not change the figure design to avoid loss of information.

5. It's hard to determine nor is it very obvious from the legend what the significance values in Figs 2 and 3 are in reference to. A clearer emphasis of this point in the legend would help.

REPLY TO REVIEWER:

We have changed the figure legends accordingly.

Reviewer #2 (Remarks to the Author):

Overall this new manuscript by Scarola et al. contains a lot of data, with well controlled experiments with appropriate statistical tests and fair conclusions accurately drawn from the data presented. Among other findings, the authors identify an OCT4 pseudogene mechanism of action and demonstrate this pseudogene is required for proper loading of the methyltransferase Suv39 to the Oct4 gene. The authors have run all the proper controls and the actual data presented in the figures are strong. Experimentally the paper is quite solid and well laid out in a logical manner. However, some more experimental details in the manuscript would be required to merit publication. Major and minor points in this regard below:

We thank reviewer for the positive comments on our manuscript; in particular that our paper consists of “well controlled experiments with appropriate statistical tests and fair conclusions accurately drawn from the data presented”. We appreciate the suggestions of the reviewer and have addressed all related issue in the revised version of the manuscript.

Major points:

1. Some of the experiments presented are fairly complex, for instance the formation of embryoid bodies. The M&M section should include more details on these experiments. Additionally, it would be nice to see images of the embryoid bodies during this experiment (either in the primary figures or supplement), so the reader can see what the cells look like @T0, and D3-10.

REPLY TO REVIEWER:

As suggested by the reviewer, we have inserted representative images on embryoid body differentiation into the **new supplementary figure 1C**. In addition, to RT-PCR data of Fig. 1E, F we now also show western blotting data in the **new supplementary figure 1B**. We further added representative movies showing contractile cardiomyocytes generated from EB bodies in the **new supplementary movies**

2. More experimental details on how actual experiments were conducted should be in the materials section. For instance, there is no description of how ChIP was conducted, how virus was packaged (packaging vectors used), how virus was concentrated or used neat, how much reagents were used for experiments such as ug DNA used in transfection and volume or MOI of virus used in infections.

REPLY TO REVIEWER:

We thank the reviewers to bring our attention on this point. We have provided additional detailed information in the supplementary information section of the revised version of the manuscript.

3. Paragraph structure could be improved. For instance:
a. The introduction is one long paragraph. This could be broken down into separate paragraphs to separate out focus areas, more typically introductions are 3-5 paragraphs setting the tone for the manuscript and giving appropriate background to appreciate the work. For example, more details about the known roles of FUS would be helpful in understanding the importance of this manuscript in context to the field.

REPLY TO REVIEWER:

We thank the reviewer for suggesting improvements of the introduction. The text was worked over and we have better separated the text into different paragraphs.

We also included a brief description of FUS in the results section of the revised version of the manuscript, **page 11, line 18 to 22**: “*In addition to transcriptional regulation, FUS has been demonstrated to be involved in DNA repair, alternative splicing, transcriptional regulation, RNA localization and stress granules (PMID: 25289647). FUS translocation events and mutations have been linked with liposarcoma and amyotrophic lateral sclerosis (ALS), respectively*”.

b. The results sections are all one paragraph with sometimes a separate thought as a single sentence paragraph. These also could be expanded to better go through the large amount of experimental data in this manuscript.

REPLY TO REVIEWER:

We have worked over the manuscript, following the suggestions of the reviewer.

Minor points:

4. The title contains a lot of acronyms that are not defined, for instance HMTase could be removed and the title would have the same meaning.

REPLY TO REVIEWER:

We have changed the title of the manuscript accordingly.

5. Gene names are sometimes capitalized and sometimes first letter in caps. This should be consistent throughout the document. For instance, the Oct4 gene is written as "OCT4" and "Oct4" in the abstract. It should be first letter caps, and preferable in italics when referring to the gene name throughout.

REPLY TO REVIEWER:

As suggested by the reviewer, in the manuscript we have used the following nomenclature (using the SHH gene as an example)

Human:

Gene: capital letters italic *SHH*

RNA: *SHH*:

Protein: SHH

Mouse:

Gene: Italic, first letter capital letter *Shh*

RNA: *Shh*

Protein: SHH

All gene-RNA-protein names have been therefore now carefully checked.

6. Paper would benefit from additional proofreading for proper use of gene names, grammar, and paragraph structure.

REPLY TO REVIEWER:

We have made our best to correct all possible mistakes

REVIEWERS' COMMENTS:

Reviewer #1 (Remarks to the Author):

In their revised manuscript, Scarola et al have addressed all my concerns and I can support their manuscript for publication. The new data and analysis provide important insight and context to their model to make it easier to understand and further increases its interest.

I find the 2 lncRNA molecules per cell not too unreasonable and may in some future study be found to reflect a dynamic cycle to OCT4/lncRNA regulation rather than a static occupancy of the promoter by these two molecules. The model proposed here for FUS is reminiscent of the concept of an "RNA chaperone" which can aid RNA-folding to allow important sequence/structures to be presented to binding partners properly. Examples can include the snRNA binding partners during splicing and the binding of sRNA's by Hfq proteins in bacteria.

I have only some minor comments that may aid the final version of the manuscript.

1 - I wanted to confirm that the labels for "empty" and "Oct4P4" in Supplemental Figure 1B were not swapped. However, it is not necessary that day 6 appear the same in western and qRT-PCR since day 10 maybe confirms the same conclusion. This reader was somewhat confused by the presence of two bands for OCT4 in the western Supp Fig 1B. The authors might provide some explanation in the figure's legend unless I've just overlooked something obvious.

2 - Page 9 line 18, the words "by murine and mouse mOct4P4 and hOCT4P3" may be a mistake.

3 - In Figure 5A, it would be helpful for the authors to label the lower band as primer dimers. Otherwise the non-specific band may be easily misinterpreted to contradict the written description in the text.

4 - Page 13 line 15, the word "exclusively" to describe FUS binding is such a strong word and seems unexpectedly out of alignment with the author's discussion elsewhere in the manuscript and in the response letter. I think the implicit limitation of "to the limit of detection" might argue counter to the idea of exclusivity. Nevertheless, the larger picture that the lncRNA is notably enriched for FUS binding is understood.

5 - I was surprised and would not have predicted that siSUV39H1 would prevent the lncRNA from binding the promoter (Figure 6H). Perhaps this again speaks to some cycling of the lncRNA through binding and leaving the promoter that allows both molecules to mutually support each other's occupancy at the site on DNA.

Reviewer #2 (Remarks to the Author):

All my major and minor points have been addressed adequately by the authors.

Rebuttal Letter

Reviewer 1

We thank reviewer 1 for having carefully studied the revised version of the manuscript. We are grateful for the comments that improve the quality of our manuscript. A detailed reply to the reviewer's comments can be found below.

REVIEWER 1:

1. I wanted to confirm that the labels for "empty" and "Oct4P4" in Supplemental Figure 1B were not swapped. However, it is not necessary that day 6 appear the same in western and qRT-PCR since day 10 maybe confirms the same conclusion. This reader was somewhat confused by the presence of two bands for OCT4 in the western Supp Fig 1B. The authors might provide some explanation in the figure's legend unless I've just overlooked something obvious.

REPLY TO REVIEWER'S COMMENT:

We thank the reviewer to highlight these issues. As suggested, we do not show data from EB D6 in the revised version of the manuscript. In addition, we indicate the unspecific OCT4 band with an asterisk (*). According changes where introduced into the respective figure legend.

REVIEWER 1

2 - Page 9 line 18, the words "by murine and mouse mOct4P4 and hOCT4P3" may be a mistake.

REPLY TO REVIEWER COMMENT:

We have corrected this mistake in the final version of our manuscript (Page 6, Line 18).

REVIEWER 1

3 - In Figure 5A, it would be helpful for the authors to label the lower band as primer dimers. Otherwise the non-specific band may be easily misinterpreted to contradict the written description in the text.

REPLY TO REVIEWER'S COMMENT:

We thank the author for this comment. We have introduced the requested change in Fig. 5a in the final version of our manuscript. We changed the figure legend accordingly.

REVIEWER 1

4 - Page 13 line 15, the word "exclusively" to describe FUS binding is such a strong word and seems unexpectedly out of alignment with the author's discussion elsewhere in the manuscript and in the response letter. I think the implicit limitation of "to the limit of detection" might argue counter to the idea of exclusivity. Nevertheless, the larger picture that the lncRNA is notably enriched for FUS binding is understood.

REPLY TO REVIEWER'S COMMENT:

We agree with the author that the word "exclusively" is too strong in this context. In order to resolve this issue, we have changed this sentence in the final version of the manuscript. We now state: "Importantly, RIP experiments using mESCs demonstrated that under our experimental conditions SUV39H1 and FUS *display binding specificity towards mOct4P4* lncRNA but not Oct4 or other_mRNAs such as Sox2, Nanog, Gapdh or Actin (Fig. 5e, f)."

This modification can be found on Page 10, Line 12-14.

REVIEWER 1

5 - I was surprised and would not have predicted that siSUV39H1 would prevent the lncRNA from binding the promoter (Figure 6H). Perhaps this again speaks to some cycling of the lncRNA through binding and leaving the promoter that allows both molecules to mutually support each other's occupancy at the site on DNA.

REPLY TO REVIEWER'S COMMENT:

We thank the reviewer for this comment. We currently explore possible mechanisms that specifically load the silencing complex to the *Oct4* promoter. Such factors may comprise additional protein factors or epigenetic information already present at the target promoter.

Reviewer 2

1 All my major and minor points have been addressed adequately by the authors.

REPLY TO REVIEWER'S COMMENT:

We thank Reviewer 2 for accepting the revised version of the manuscript.